# Identification of human glucocorticoid response markers using integrated multi-omic analysis from a randomized crossover trial

Dimitrios Chantzichristos[1,2]*, Per-Arne Svensson[3,4], Terence Garner[5], Camilla AM Glad[1,2], Brian R Walker[6,7], Ragnhildur Bergthorsdottir[1,2], Oskar Ragnarsson[1,2], Penelope Trimpou[1,2], Roland H Stimson[7], Stina W Borresen[8,9], Ulla Feldt-Rasmussen[8,9], Per-Anders Jansson[10], Stanko Skrtic[1,11], Adam Stevens[5†], Gudmundur Johannsson[1,2†]

[1]Department of Internal Medicine and Clinical Nutrition, Institute of Medicine at Sahlgrenska Academy, University of Gothenburg, Gothenburg, Sweden; [2]Endocrinology, Diabetology and Metabolism, Sahlgrenska University Hospital, Gothenburg, Sweden; [3]Department of Molecular and Clinical Medicine, Institute of Medicine at Sahlgrenska Academy, University of Gothenburg, Gothenburg, Sweden; [4]Institute of Health and Care Sciences, Sahlgrenska Academy, University of Gothenburg, Gothenburg, Sweden; [5]Division of Developmental Biology & Medicine, Faculty of Biology, Medicine and Health, University of Manchester, Manchester, United Kingdom; [6]Clinical and Translational Research Institute, Newcastle University, Newcastle upon Tyne, United Kingdom; [7]BHF/University Centre for Cardiovascular Science, University of Edinburgh, Edinburgh, United Kingdom; [8]Department of Medical Endocrinology and Metabolism, Copenhagen University Hospital, Copenhagen, Denmark; [9]Department of Clinical Medicine, Faculty of Health and Medical Sciences, University of Copenhagen, Copenhagen, Denmark; [10]Wallenberg Laboratory, Department of Molecular and Clinical Medicine, Institute of Medicine at Sahlgrenska Academy, University of Gothenburg, Gothenburg, Sweden; [11]Innovation Strategies and External Liaison, Pharmaceutical Technologies and Development, Gothenburg, Sweden

*For correspondence:
dimitrios.chantzichristos@gu.se

†These authors contributed equally to this work

## Abstract

**Background:** Glucocorticoids are among the most commonly prescribed drugs, but there is no biomarker that can quantify their action. The aim of the study was to identify and validate circulating biomarkers of glucocorticoid action.

**Methods:** In a randomized, crossover, single-blind, discovery study, 10 subjects with primary adrenal insufficiency (and no other endocrinopathies) were admitted at the in-patient clinic and studied during physiological glucocorticoid exposure and withdrawal. A randomization plan before the first intervention was used. Besides mild physical and/or mental fatigue and salt craving, no serious adverse events were observed. The transcriptome in peripheral blood mononuclear cells and adipose tissue, plasma miRNAomic, and serum metabolomics were compared between the interventions using integrated multi-omic analysis.

**Results:** We identified a transcriptomic profile derived from two tissues and a multi-omic cluster, both predictive of glucocorticoid exposure. A microRNA (miR-122-5p) that was correlated with

genes and metabolites regulated by glucocorticoid exposure was identified (p=0.009) and replicated in independent studies with varying glucocorticoid exposure (0.01 ≤ p≤0.05).

**Conclusions:** We have generated results that construct the basis for successful discovery of biomarker(s) to measure effects of glucocorticoids, allowing strategies to individualize and optimize glucocorticoid therapy, and shedding light on disease etiology related to unphysiological glucocorticoid exposure, such as in cardiovascular disease and obesity.

**Funding:** The Swedish Research Council (Grant 2015-02561 and 2019-01112); The Swedish federal government under the LUA/ALF agreement (Grant ALFGBG-719531); The Swedish Endocrinology Association; The Gothenburg Medical Society; Wellcome Trust; The Medical Research Council, UK; The Chief Scientist Office, UK; The Eva Madura's Foundation; The Research Foundation of Copenhagen University Hospital; and The Danish Rheumatism Association.
**Clinical trial number:** NCT02152553.

## Introduction

Glucocorticoids (GCs) have a key role in the metabolic, vascular, and immunological response to stress (*Cain and Cidlowski, 2017*; *Oster et al., 2017*). GC secretion from the adrenal gland is under tight dynamic control by the hypothalamic–pituitary–adrenal axis and is regulated in a classic circadian pattern (*Cain and Cidlowski, 2017*; *Oster et al., 2017*). Most actions of GCs are mediated by the ubiquitously expressed GC receptor (*Cain and Cidlowski, 2017*; *Oster et al., 2017*). The tissue-specific effects of GCs are regulated by many local factors, including pre-receptor metabolism of GCs and the interaction of the GC receptor with tissue-specific transcription factors, or through non-genomic mechanisms (*Cain and Cidlowski, 2017*; *Oster et al., 2017*). As a result of this complexity, circulating levels of cortisol relate poorly to tissue action of cortisol, and serum cortisol therefore has limited value as a biomarker for GC action (*Karssen et al., 2001*).

GCs are among the most commonly prescribed drugs, and GC treatment remains a cornerstone in the management of many rheumatic and inflammatory diseases despite the introduction of modern disease-modifying antirheumatic drugs and biological immunomodulatory treatment (*Smolen et al., 2017*). GC replacement is essential for survival in patients with various forms of adrenal insufficiency (*Johannsson et al., 2015*). However, metabolic and other side effects of GC treatment or replacement are common (*Björnsdottir et al., 2011*; *Fardet et al., 2012*), indicating that current methods to monitor their action and tailor their treatment are inadequate. Unphysiological GC exposure has been implicated in the etiology of several common diseases such as type 2 diabetes mellitus, hypertension, abdominal obesity, and cardiovascular disease (*Ragnarsson et al., 2019*).

Against this background, it is highly desirable to be able to measure and quantify GC action as this might be useful to refine current GC therapy. Biomarkers of GC action will also provide potential mechanistic understanding for the role of GCs in the etiology of many common diseases. Previous attempts to identify biomarkers using metabolomics have identified circulating metabolites associated with GC exposure (*Alwashih et al., 2017a*; *Alwashih et al., 2017b*). Integrated multi-omic analysis provides increased robustness over analysis of individual 'omic data sets (*Ideker et al., 2011*). In particular, the identification of groups within one 'omic 'layer' with shared co-regulation within another 'omic layer implies a functional relationship that can be used both to assess the mechanistical relevance and to support the identification of biomarkers (*Karczewski and Snyder, 2018*; *Misra et al., 2018*).

The aim of this exploratory study was to define multi-omic patterns derived from independent tissues related to GC action and to use these patterns to search for clinically applicable circulating biomarkers of GC action. Subjects with primary adrenal insufficiency, Addison's disease, lack GC production from the adrenal cortex and can therefore be considered a human GC 'knock-down' model (*Figure 1A*). An experimental study design including subjects with Addison's disease, standardizing for diurnal variation and food intake, allowed a within-individual comparison between physiological GC exposure and GC withdrawal (*Figure 1B*). A multi-omic analysis strategy combining data from gene expression in circulation (peripheral blood mononuclear cells [PBMCs]) and an important metabolic tissue, adipose tissue, integrated with circulating microRNAs (miRNAs) and metabolites was used to identify putative biomarkers. The strongest putative biomarkers were then replicated in independent study groups with different GC exposure.

**eLife digest** Several diseases, including asthma, arthritis, some skin conditions, and cancer, are treated with medications called glucocorticoids, which are synthetic versions of human hormones. These drugs are also used to treat people with a condition call adrenal insufficiency who do not produce enough of an important hormone called cortisol. Use of glucocorticoids is very common, the proportion of people in a given country taking them can range from 0.5% to 21% of the population depending on the duration of the treatment. But, like any medication, glucocorticoids have both benefits and risks: people who take glucocorticoids for a long time have an increased risk of diabetes, obesity, cardiovascular disease, and death.

Because of the risks associated with taking glucocorticoids, it is very important for physicians to tailor the dose to each patient's needs. Doing this can be tricky, because the levels of glucocorticoids in a patient's blood are not a good indicator of the medication's activity in the body. A test that can accurately measure the glucocorticoid activity could help physicians personalize treatment and reduce harmful side effects.

As a first step towards developing such a test, Chantzichristos et al. identified a potential way to measure glucocorticoid activity in patient's blood. In the experiments, blood samples were collected from ten patients with adrenal insufficiency both when they were on no medication, and when they were taking a glucocorticoid to replace their missing hormones. Next, the blood samples were analyzed to determine which genes were turned on and off in each patient with and without the medication. They also compared small molecules in the blood called metabolites and tiny pieces of genetic material called microRNAs that turn genes on and off.

The experiments revealed networks of genes, metabolites, and microRNAs that are associated with glucocorticoid activity, and one microRNA called miR-122-5p stood out as a potential way to measure glucocorticoid activity. To verify this microRNA's usefulness, Chantzichristos et al. looked at levels of miR-122-5p in people participating in three other studies and confirmed that it was a good indicator of the glucocorticoid activity.

More research is needed to confirm Chantzichristos et al.'s findings and to develop a test that can be used by physicians to measure glucocorticoid activity. The microRNA identified, miR-122-5p, has been previously linked to diabetes, so studying it further may also help scientists understand how taking glucocorticoids may increase the risk of developing diabetes and related diseases.

## Results

### Clinical experimental study

#### Patient characteristics

Eleven subjects with well-defined Addison's disease and no other endocrinopathies were recruited and included in the study between September 2013 and September 2015. One subject discontinued the study after randomization and before the first intervention because of persistent orthostatic hypotension. Ten subjects (four women with three of them post-menopausal) with a median age of 50 years (range, 25–57) and a median disease duration of 23.5 years (range, 1–33) completed all aspects of the study between May 2014 and October 2015. The median daily replacement dose of hydrocortisone (HC) prior to the study was 30 mg (range, 20–30), and 9 out of 10 subjects had treatment with fludrocortisone (mineralocorticoid) at a median daily dose of 0.1 mg (range, 0.1–0.2).

#### Clinical and biochemical outcomes

The main time points for sample collection in each intervention were at 9 AM on the first intervention day ('before start') and at 7 AM on the second intervention day ('morning') (*Figure 1B*). The subjects' last ordinary oral HC dose was administered the day before admission to the study unit.

Infusion of HC mixed with isotonic saline ('GC exposure') had no effect on systolic and diastolic blood pressure, body weight, serum sodium and potassium, or plasma glucose concentrations compared to the same amount of isotonic saline infusion alone ('GC withdrawal') (*Table 1*). HC and saline infusion achieved the intended differences in GC exposure. Both median morning serum cortisol and cortisone during the HC infusion were within the physiological range (298 and 81.2 nmol/L,

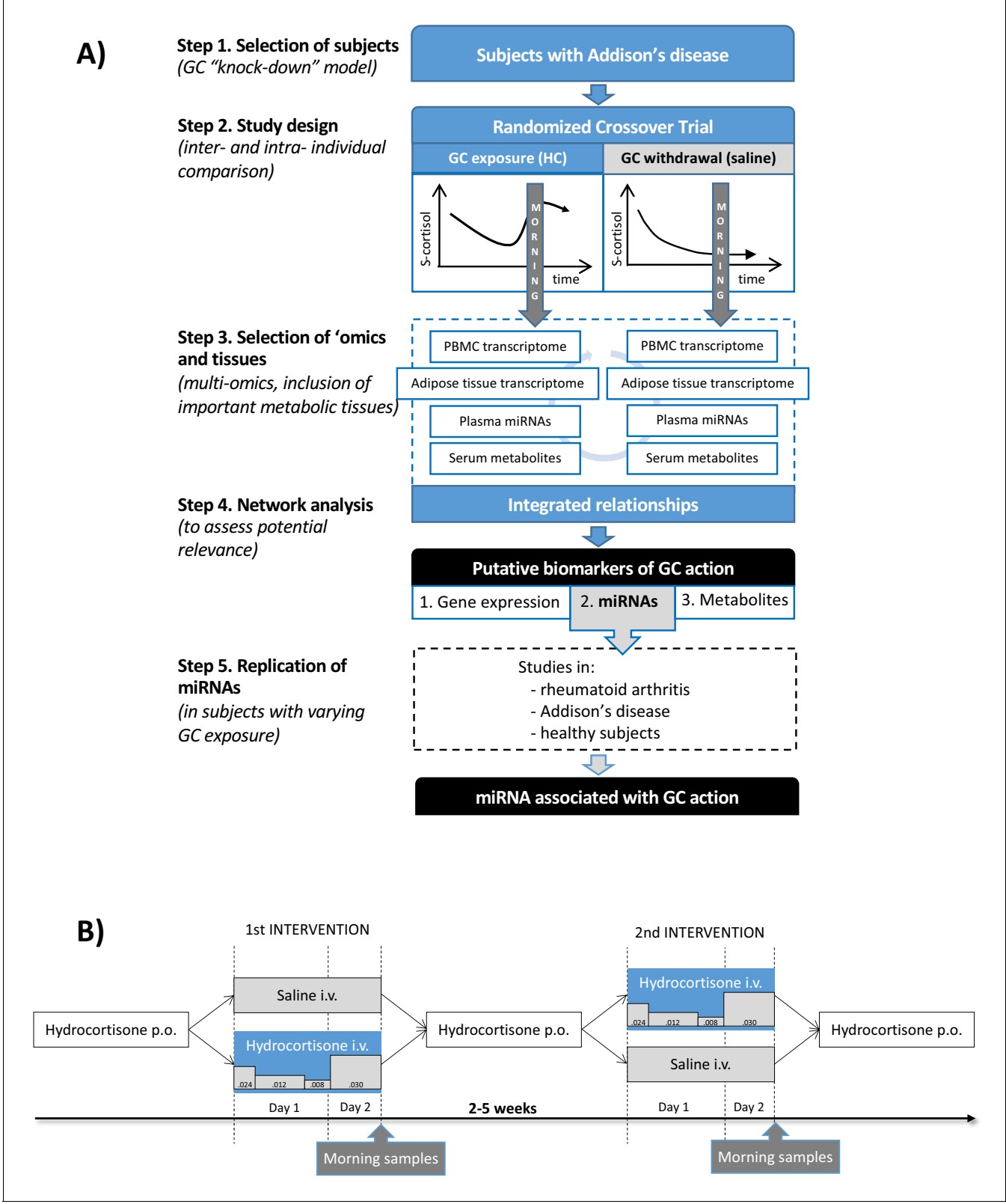

**Figure 1.** Clinical and analytical part of the exploratory study and the replication step. (**A**) Subjects with Addison's disease (primary adrenal insufficiency, step 1) were studied in a random order during both physiological glucocorticoid (GC) exposure and GC withdrawal (step 2). Transcriptomics (whole-genome expression) in peripheral blood mononuclear cells (PBMCs) and adipose tissue (n = 28,869,869 genes), plasma miRNAomics (n = 252), and serum metabolomics in morning samples were analyzed (n = 164) (step 3). Integration of the multi-omic data derived a

*Figure 1 continued on next page*

*Figure 1 continued*

network including gene expression (derived from two independent tissues), microRNAs (miRNAs), and metabolites that were statistically differentiated between the two interventions (step 4). The miRNA findings, because of their centrality in the network, were replicated in subjects with different GC exposures (within the physiological range) from three independent studies (step 5). (**B**) Subjects with Addison's disease (primary adrenal insufficiency) received in a random order intravenous (i.v.) hydrocortisone (HC) infusion mixed in 0.9% saline in a circadian pattern (physiological GC exposure) or the same volume of 0.9% saline alone (GC withdrawal) during 22 hr starting at 9 AM more than 2 weeks apart. During the GC exposure, HC(Solu-Cortef) was administered at a dose of 0.024 mg/kg/hr between 9 AM and 12 PM (first day), 0.012 mg/kg/hr between 12 PM and 8 PM (first day), 0.008 mg/kg/hr between 8 PM and 12 AM (first day), and 0.030 mg/kg/hr between 12 AM and 7 AM (second day). Samples for the 'omics analyses were collected at 7 AM on day 2 of the intervention (morning samples). p.o.: oral.

respectively) and markedly lower during the saline infusion (44.4 and 42 nmol/L, respectively, both p<0.001) (*Figure 2*). Serum cortisol and cortisone were detected in all subjects' morning samples during the saline infusion, but both overnight (between 12 AM and 7 AM) urinary cortisol and cortisone excretion were below the limit of detection. Both HC and saline infusions were well-tolerated, and no serious adverse events were observed. Three subjects reported mild physical and/or mental fatigue, and one subject reported mild salt craving during the GC withdrawal period.

**Table 1.** Clinical and biochemical tests assessed or collected immediately before each intervention and at 7 AM on the second intervention day during both interventions ($n$ = 10).

| | | 9 AM before start of intervention (first intervention day) | | 7 AM (second intervention day) | |
|---|---|---|---|---|---|
| | | Median (IQR) | p-Value | Median (IQR) | p-Value |
| S-cortisol (nmol/L) | HC | 43.2 (38.0–55.1) | 0.36 | 298 (228–359) | <0.001 |
| | Saline | 46.7 (43.1–61.7) | | 44.4 (36.8–52.5) | |
| Overnight U-free cortisol (µg/7 hr)*,[†] | HC | – | | 678 (459–814) | <0.001 |
| | Saline | – | | <0.01[‡] | |
| S-cortisone (nmol/L) | HC | 40.7 (29.5–49.2) | 0.76 | 81.2 (60.5–94.9) | <0.001 |
| | Saline | 42.1 (29.7–50.4) | | 42.0 (28.9–47.3) | |
| Overnight U-free cortisone (µg/7 hr)*,[†] | HC | – | | 136 (106–151) | <0.001 |
| | Saline | – | | <0.01[‡] | |
| SBP (mmHg) | HC | 122 (111–134) | 0.88 | 123 (107–139) | 0.65 |
| | Saline | 127 (112–131) | | 124 (101–137) | |
| DBP (mmHg) | HC | 76 (63–83) | 0.88 | 69 (59–76) | 0.55 |
| | Saline | 72 (67–80) | | 65 (60–70) | |
| S-sodium (mmol/L) | HC | 141 (139–142) | 0.85 | 140 (138–141) | 0.97 |
| | Saline | 141 (138–142) | | 140 (138–141) | |
| S-potassium (mmol/L) | HC | 4.4 (4.3–4.7) | 0.27 | 4.4 (4.3–4.8) | 0.26 |
| | Saline | 4.2 (4.0–4.5) | | 4.4 (4.3–4.5) | |
| P-glucose (mmol/L) | HC | 5.0 (4.5–5.3) | 0.79 | 5.6 (5.2–5.8) | 0.08 |
| | Saline | 5.0 (4.6–5.1) | | 5.2 (5.2–5.4) | |
| Body weight (kg) | HC | 72.2 (69.6–77.3) | 0.53 | 72.8 (69.3–77.7) | 0.63 |
| | Saline | 73.5 (70.5–79.5) | | 74.0 (70.5–79.2) | |

*Overnight U-free cortisol and cortisone were collected from midnight to morning (12 AM to 7 AM) during physiological GC exposure (HC infusion) and GC withdrawal (saline infusion).

[†]One of the 10 subjects was not included in the analysis because of a problem during sample collection.

[‡]Below the limit of detection.

DBP: diastolic blood pressure; GC: glucocorticoid; HC: hydrocortisone; IQR: interquartile range; P: plasma; S: serum; SBP: systolic blood pressure; U: urinary.

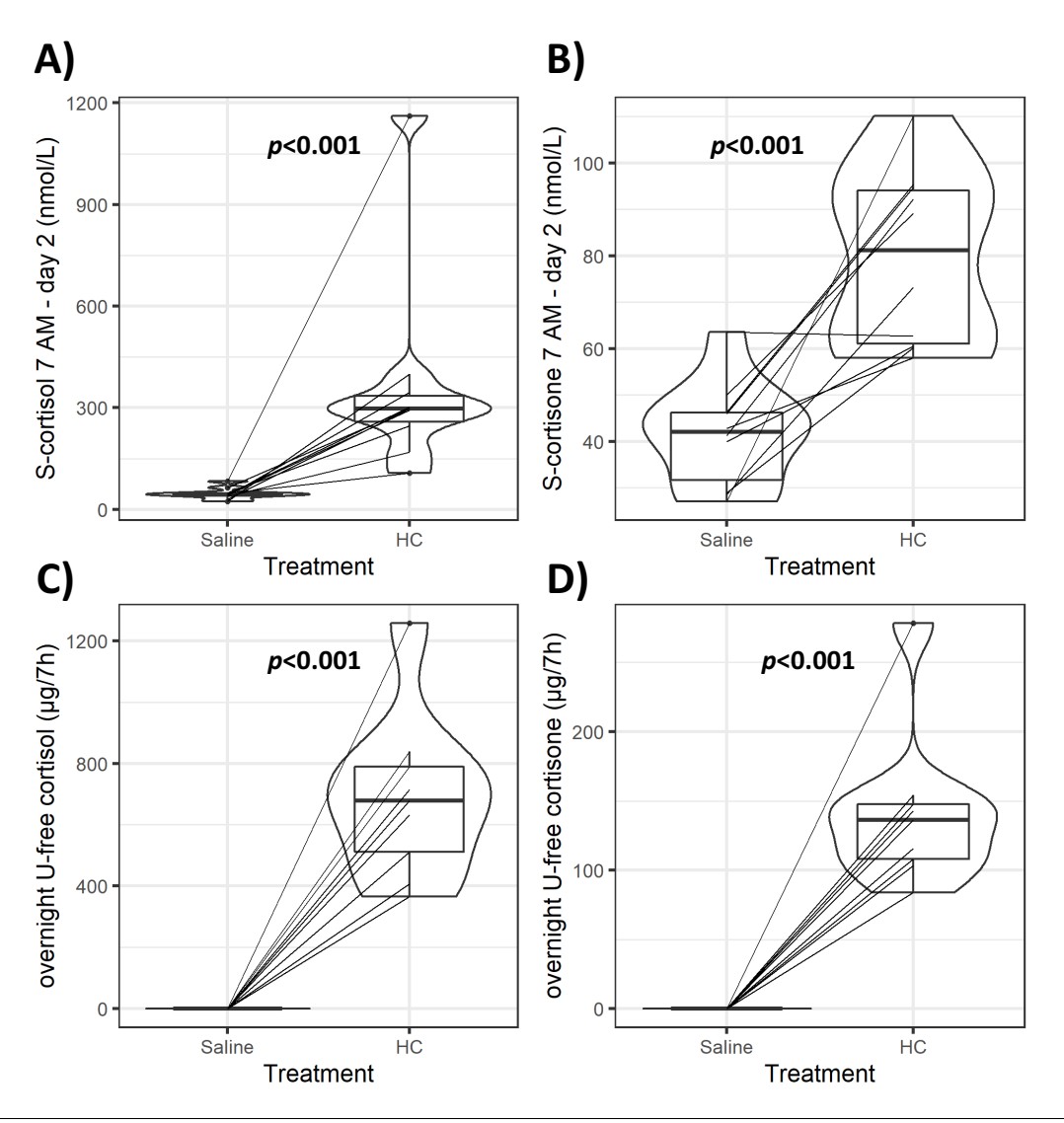

**Figure 2.** Violin plots of serum and urinary glucocorticoids (GCs) during GC exposure and withdrawal. Individual data and changes for morning (7 AM) serum cortisol and cortisone (**A**, **B**), and overnight (12 AM to 7 AM) urinary-free (U-free) cortisol and cortisone (**C**, **D**) from both interventions. Boxes represent interquartile range (IQR); whiskers mark spread of points within 1.5 times IQR; violins demonstrate distribution of results with the maximum width representing the highest density within each violin. Difference of median values between interventions is presented with p-values. HC: hydrocortisone.

## Differentially regulated 'omic elements associated with response to GCs

Similarity network fusion (SNF) was used to demonstrate overall similarity between subjects across and between 'omic layers, prior to analysis (Appendix 1 and *Appendix 1—figure 1*). Differential gene expression was associated with GC response in both PBMC and adipose tissue (Appendix 1). Differential expression of metabolites and miRNA was identified in blood in relation to GC response (Appendix 1). Differentially expressed 'omic elements (DEOEs) are presented in *Table 2* and *Supplementary file 1a–d*. All DEOEs were used for integrated analysis, and false discovery rate (FDR)-corrected DEOEs were used for all other analyses (*Table 2*). DEOEs from the PBMC and adipose tissue transcriptomes were shown to have limited overlap in response to GC but were enriched for shared pathways, revealing an overlap that indicated shared mechanism in relation to GC exposure (Appendix 2 and *Appendix 2—figure 3*).

**Table 2.** Summary of differentially expressed 'omic elements in association with response to glucocorticoids.

| 'Omic data set | Total number of 'omic elements | Number of significant elements (p<0.05) | Number of significant elements (FDR < 0.05) |
| --- | --- | --- | --- |
| PBMC transcriptome | 28,869 | 4426 | 3997 |
| Adipose tissue transcriptome | 28,869 | 3520 | 3115 |
| Plasma miRNAome | 252 | 9 | 9 |
| Serum metabolome | 164 | 38 | 14 |

FDR: false discovery rate; miRNA: microRNA; PBMC: peripheral blood mononuclear cell.

We assessed the impact of differential expression on the entire interactome to aid in the identification of similar GC-related function. Interactome network models were generated using differentially expressed genes (DEGs) from both the PBMC transcriptome and the adipose tissue transcriptome. These were shown to be consistent with one another (Appendix 2 and *Appendix 2—figures 1* and *2*) despite the limited overlap of DEGs. GC-responsive genes were shown to have higher connectivity in the human interactome than expected by chance, demonstrated using 10,000 permutations of this network model (Appendix 2).

## Integration of PBMC and adipose tissue transcriptomes with plasma miRNAomic and serum metabolomic data

Hypernetworks are network structures where edges are not restricted to defining a relationship between two nodes but may be shared between many nodes. As such, these structures can be used to describe complex relationships that link multiple elements. Hypernetworks also allow for the same pair of nodes to be connected by multiple edges. This means that relationships between nodes can be ranked by the number of edges shared between them. Hypernetworks allow for the summary of correlation matrices, compressing the high-dimensional relationships between data points (transcripts/miRNA/metabolites) into a single metric of similarity. Hypernetworks facilitate integration of 'omic data and can be used to define strongly associated elements. Elements with large numbers of shared edges are more similar and likely to be of functional relevance; clustering allows refinement of large 'omic data sets to highly associated elements (*Figure 3A, B*). Hypernetworks are robust to random error and act to filter out false-positive correlations as these will not have a uniform pattern of correlation across all 'omic elements.

To assess similarity, we defined the correlation coefficient between each differentially expressed 'omic measurement and assessed as 'present' in the network model those correlations with an *r*-value >|1.5| standard deviations (sd). Edges were defined as PBMC transcripts with shared correlations, for example, two PBMC transcripts that are both correlated with the same three metabolites are connected by three edges. We summarized the shared correlations as a measure of similarity between each pair of GC-responsive PBMC transcripts, counting correlations across the other 'omic data sets (*Figure 3—figure supplement 1*). The greatest number of correlations shared was between PBMC and adipose tissue transcriptome (525 genes, *Figure 3C*), reinforcing the observation that, while the gene-level overlap of differential expression was limited, common pathways are active in both tissues related to GC action, which involve similar networks of co-expressed genes. The rank order of the number of correlations shared with the GC-responsive PBMC transcriptome was adipose tissue transcriptome > plasma miRNAome > serum metabolome, and this was confirmed both by comparison of the heat maps (*Figure 3—figure supplement 1*) and by a Venn diagram (*Figure 3D*). The Venn diagram also reveals a strong correspondence between the serum metabolome and both PBMC and adipose tissue transcriptomes.

## Identification and validation of a shared transcriptomic profile in both PBMCs and adipose tissue predicting GC response

Robustness testing was performed in which hypernetworks were generated to model dissimilarity based on the absence of correlations with PBMC transcripts. Any genes that were highlighted by these hypernetworks were removed from the downstream predictive analysis. Using this approach, we defined 271 of 965 PBMC transcripts with maximum predictive potential. This set of genes

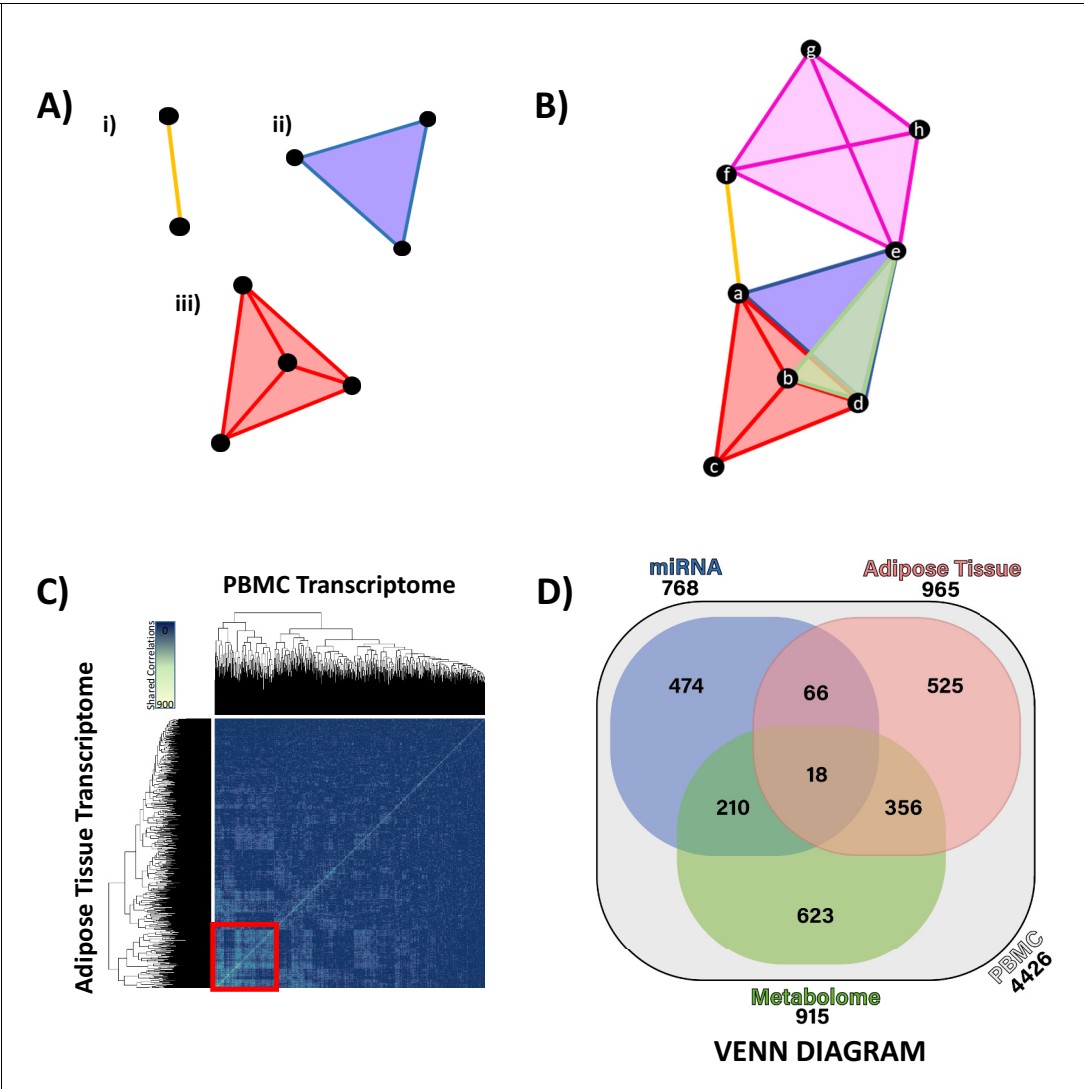

**Figure 3.** Hypernetwork analyses of integrated 'omic data. (**A**) Hypernetworks differ from traditional networks in that edges can connect more than two nodes. Nodes are represented by black circles, edges by colored lines and surfaces. This demonstration shows how one edge can connect (i) two nodes as a one-dimensional line, (ii) three nodes as a two-dimensional surface, and (iii) four nodes as a three-dimensional structure. Hypernetworks of 'omic data can have edges shared between hundreds of nodes. (**B**) Hypernetwork diagram illustrating how a pair of nodes (a–h) can be connected by more than one edge. In this example, nodes *e* and *d* share two edges, as do *b* and *d*. (**C**) A hypernetwork plotted as a heat map can be used to investigate clustering of blood peripheral blood mononuclear cell (PBMC) transcripts, based on correlation to, for example, adipose tissue transcriptome. A central cluster, defined using hierarchical clustering, groups PBMC transcripts based on high numbers of shared edges (red square, *n* = 965). This approach was applied to define groups of PBMC transcripts with similar profiles when correlated against each other 'omic layer. (**D**) Gene probe level overlaps between PBMC transcriptome clusters identified by hypernetwork shared with the other 'omic data sets. PBMC transcriptomic changes are correlated with changes in miRNAome, adipose tissue transcriptome, and metabolome (gas chromatography-mass spectrometry and liquid chromatography-mass spectrometry overlaps combined); overlaps are common PBMC transcripts with correlation to the 'omic data sets. Values in brackets represent the size of PBMC transcriptomic clusters drawn from all differentially expressed PBMC transcripts (*n* = 4426, p<0.05). Data demonstrates a fundamental relationship in glucocorticoid response between PBMC and adipose tissue (965 genes) and reinforces the presence of common pathways in these two independent tissues.

The online version of this article includes the following figure supplement(s) for figure 3:

**Figure supplement 1.** Hypernetwork heat maps of integrated 'omic data.

**Figure supplement 2.** Transcriptome quality control measures.

**Figure supplement 3.** Metabolome quality control measures.

**Figure supplement 4.** Chi-squared distance distribution from a range of dichotomization thresholds.

perfectly classified the HC- and saline-treated groups using partial least squares discriminant analysis (PLS-DA) (*Figure 4A*). We identified variables of importance using Random Forest and modeled the background experimental noise using permutation analysis (BORUTA) (*Figure 4B*). This identified a set of 59 genes as variables of importance with fold changes in the same directions in both transcriptomic data sets that perfectly classified HC from saline treatment (*Supplementary file 1e*). Nine of these genes were significantly differentially expressed in both PBMC and adipose tissue transcriptomes (*Figure 4C*), and, of these nine genes, six were associated with GC response via gene ontology (*IL18RAP*, *JAK2*, *MTSS1*, *RIN2*, *KIF1B*, and *BCL9L*) (*Figure 4D*). The gene set (*n* = 59) that we identified, which classified both PBMC and adipose tissue transcriptomes in relation to GC exposure, was validated (area under the curve [AUC] 0.70–0.96) by further testing in five other previous studies of GC action by other research groups in cellular models (*Table 3*). Further robustness of the random forest observations was provided by demonstrating that the minimal depth at which the variables of importance became active in prediction was small (*Figure 4—figure supplement 1*).

## Integration of circulating 'omic data sets leads to miRNA and metabolite markers of GC action

We further examined interactions between the circulating 'omics data associated with GC exposure (*Figure 3D*). All of the circulating 'omics data was combined to form a correlation matrix and hierarchical clustering used to identify 'omic data points with similar correlation (*Figure 5—figure supplement 1*). Eleven clusters including transcriptomic, miRNAomic, and metabolomic data were identified, and these clusters were shown to have enrichment within the interactome network model (*Supplementary file 1f* and Appendix 2).

We then quantified the number of correlations between all the circulating 'omic data associated with GC exposure (*n* = 336) using a hypernetwork. This approach was used to define a group of highly connected multi-omic elements with a relationship to GC exposure (*Figure 5A*).

A hypernetwork model of the core group of 139 highly connected elements was generated (*Figure 5B*). *DCK* was the only gene shared with the GC-dependent adipose tissue transcriptome that also had predictive value (highlighted with a red square in *Figure 5B*). Deletion of the *DCK* gene region has been shown to be associated with increased sensitivity to GCs (*Malani et al., 2017*), an observation in alignment with the reduction in expression we found in both PBMC and adipose tissue transcriptomes in association with GC exposure (*Figure 4C*).

The hypernetwork model (*Figure 5B*) also highlighted a range of related miRNAs and metabolites. A hierarchical model of modules within the network was assessed using the measure of network centrality (*Figure 5C*). These modules revealed multi-omic relationships and demonstrated that miR-122-5p was the only miRNA present in higher order modules as measured by network centrality. miR-122-5p was correlated with cortisol exposure and the expression of *FKBP5*, a regulator of GC sensitivity (cluster 11 in *Figure 5—figure supplement 1* and *Supplementary file 1f*).

Targeted replication of the plasma miR-122-5p fold change from the experimental study in subjects with Addison's disease using an independent RNA separation procedure showed a marked down-regulation of miR-122-5p by increased GC exposure (p=0.009) (*Figure 6*). Two subjects did not show this miR-122-5p response, one man (disease duration 2 years, body mass index [BMI] 23.8 kg/m$^2$; hydrocortisone 20 mg daily, fludrocortisone 0.1 mg daily) and one woman (disease duration 23 years; BMI 28.1 kg/m$^2$; hydrocortisone 30 mg daily, fludrocortisone 0.2 mg daily) who both experienced mild mental fatigue during GC withdrawal.

## Replication of miRNA findings in independent study groups

Based on (i) the functional association of a circulating miRNA with gene expression and metabolomics, and (ii) the correlation between the PBMC transcriptome and plasma miRNAome (*Figure 3D*), a targeted replication of the plasma miRNA findings was conducted using an independent RNA separation procedure. Twelve miRNAs were re-analyzed in the current study and in three other independent studies including subjects with different GC exposures: (i) in 60 subjects with rheumatoid arthritis with and without tertiary adrenal insufficiency after a short-term stop in their GC treatment (low vs. physiological GC exposure, respectively) (*Borresen et al., 2017*); (ii) in 20 subjects with Addison's disease receiving HC replacement therapy and in 20 matched healthy control subjects

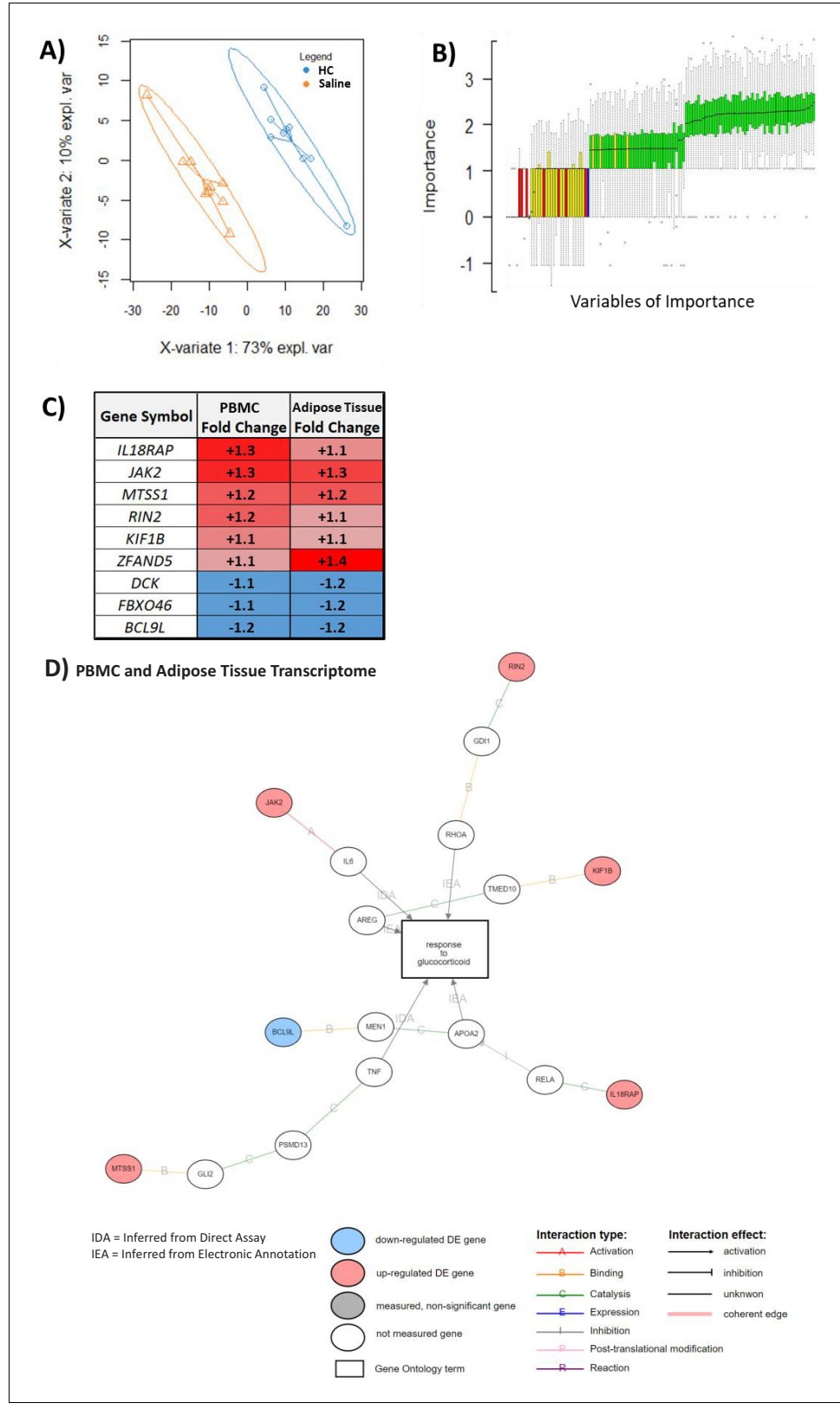

**Figure 4.** An overlapping gene set in peripheral blood mononuclear cell (PBMC) and adipose tissue transcriptome can be used to classify glucocorticoid (GC) response. These analyses were performed to depict the common predictive genes in PBMCs and adipose tissue. (**A**) Partial least squares discriminant analysis (PLSDA) showing complete separation of hydrocortisone (HC) infusion (GC exposure, blue points) from saline infusion (GC

*Figure 4 continued on next page*

*Figure 4 continued*

withdrawal, orange points) using 271 of 965 PBMC transcripts confirmed as robust in the hypernetwork by analysis of dissimilarity. X-variates 1 and 2: PLSDA components; expl. var: explained variance. (B) BORUTA feature selection identifies variables (genes) of importance in classification using a Random Forest approach to model experimental background noise (green: confirmed classification; yellow: tentative classification: red: rejected classification; blue: 'shadow' variable modeling experimental noise). Of 271 transcripts initially used, 59 were identified as important (confirmed [green] or tentative [yellow]) in separating GC exposure from GC withdrawal, as well as having the same direction fold change in both PBMC and adipose tissue transcriptomic data sets. (C) Predictive genes that are significantly differentially expressed between GC exposure and GC withdrawal in both PBMC and adipose tissue transcriptomes and display fold change in the same direction in both tissues (*n* = 9). (D) Association of predictive genes (six out of nine) with GC response through gene ontology. Data demonstrates the presence of a robust transcriptomic profile predicting GC response in two independent tissues.

The online version of this article includes the following figure supplement(s) for figure 4:

**Figure supplement 1.** Random forest distribution of minimal depth.

---

(low vs. physiological GC exposure, respectively) (*Bergthorsdottir et al., 2017*); and (iii) acute low, medium, and excessive GC exposure in 20 healthy subjects (*Stimson et al., 2017*).

From this analysis, miR-122-5p was significantly associated with different GC exposure in all studies (*Figure 7A– D*). The expression of miR-122-5p was higher in subjects with rheumatoid arthritis and reduced GC exposure due to tertiary adrenal insufficiency (*Figure 7A*), and subjects with Addison's disease had higher expression of miR-122-5p than healthy matched controls (*Figure 7B*). In the experimental study in healthy subjects, the expression of miR-122-5p was increased both after low and excessive high GC exposure compared to medium GC exposure at both high and low insulin levels (*Figure 7C, D*, respectively). The other 11 miRNAs (including miR-425-3p) did not show a relationship with GC exposure in the three replication studies.

**Table 3.** Validation of the predictive genes from the current exploratory study against previous studies examining GC response in cellular systems.

| Study title | GEO # | PMID | N | AUC (95% CI) | OOB* AUC[†] | OOB* error rate[‡] (%) |
|---|---|---|---|---|---|---|
| Dexamethasone effect on epidermal keratinocytes in vitro | GSE26487 | 17095510 (*Stojadinovic et al., 2007*) | 20 | 0.70 (0.51–0.89) | 0.80 | 30 |
| Dexamethasone effect on GC-resistant and -sensitive lymphoblastic leukemia cell lines | GSE22152 | 21092265 (*Carlet et al., 2010*) | 24 | 0.71 (0.52–0.90) | 0.78 | 29 |
| In vivo GC effect on non-leukemic peripheral blood lymphocytes | GSE22779 | 21092265 (*Carlet et al., 2010*) | 16 | 0.88 (0.63–1.0) | 0.96 | 6 |
| Osteosarcoma cell line response to activation of specific GC receptor alpha isoforms | GSE6711 | 17682054 (*Lu et al., 2007*) 22174376 (*Jewell et al., 2012*) | 60 | 0.96 (0.89–1.0) | 0.99 | 3 |
| GC effect on lens epithelial cells | GSE3040 | 16319822 (*Gupta et al., 2005*) | 12 | 0.83 (0.63–1.0) | 0.72 | 17 |

*OOB data, the bootstrapping approach of Random Forest, ensures that every tree is built using ~63% of the available data, leaving ~ 37% that can be used for a validation test.

[†]AUC up to 0.96 demonstrates a high probability of correctly classifying a randomly selected sample from each study.

[‡]OOB error rate = prediction error using the OOB validation data.

The gene set that classified both PBMC and adipose tissue transcriptomes in relation to GC exposure with fold change in the same direction (see *Figure 4B* – 59 genes) was validated by further testing in five other publicly available studies of GC action in cellular systems.

AUC: area under the curve of the receiver operating characteristic; CI: confidence interval; GC: glucocorticoid; GEO: Gene Expression Omnibus; GEO #: study number deposited with GEO; *N*: study number size; OOB: out-of-bag; PBMC: peripheral blood mononuclear cell; PMID: PubMed ID number of the manuscript describing the data.

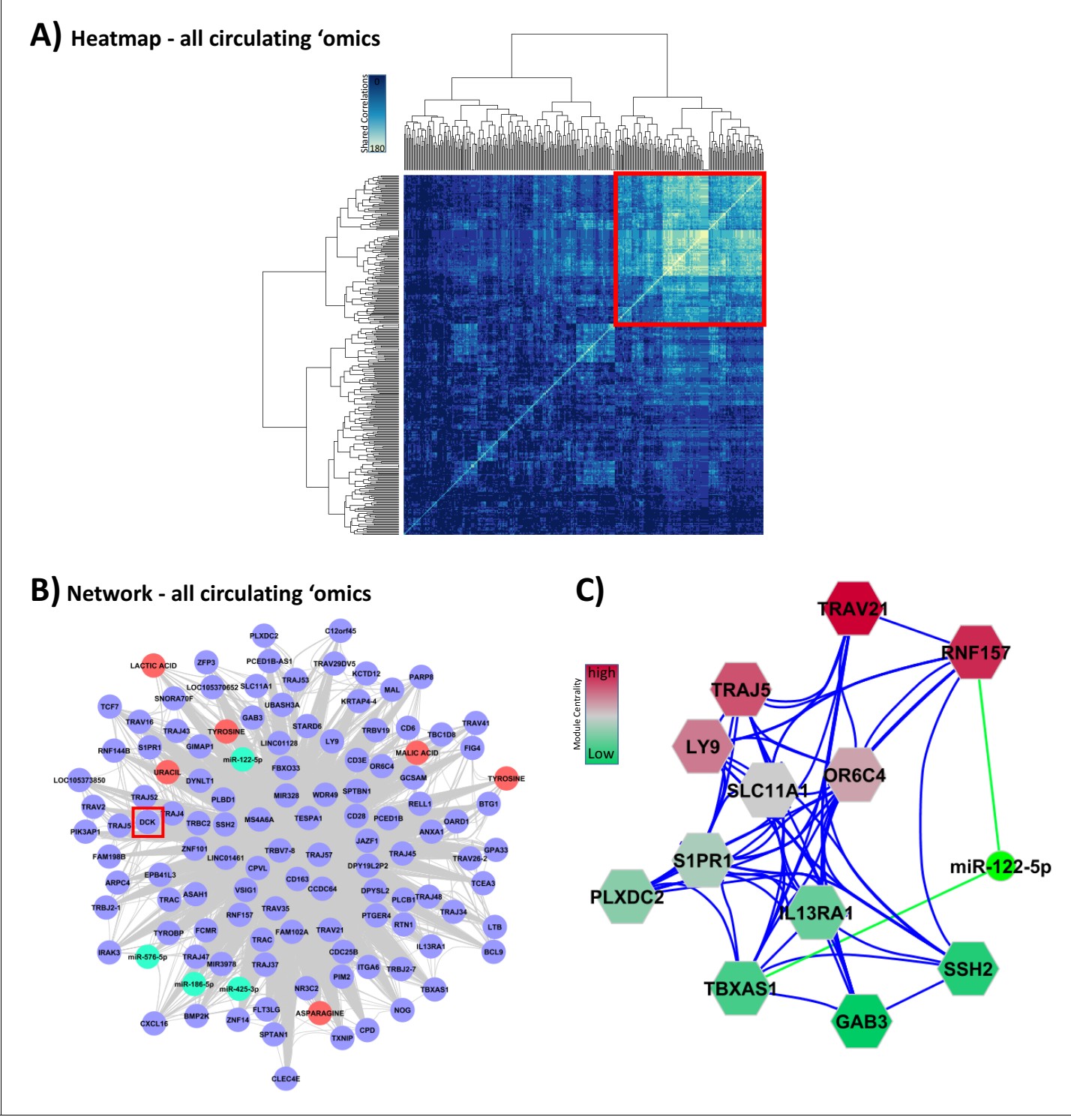

**Figure 5.** Integration of all circulating 'omic data sets associated with glucocorticoid (GC) response. These analyses were performed to lead to putative biomarkers of GC action. (**A**) Hypernetwork summary heat map of shared correlations between all circulating 'omic elements (peripheral blood mononuclear cell transcriptome, plasma microRNA [miRNA], serum metabolome; $n$ = 336) with differential expression between GC exposure and GC withdrawal. (**B**) Network representation of central cluster (red square in **A** [$n$ = 139], of which 120 map to genes/miRNA/metabolites). Blue circles: genes with differential expression; red circle: differentially expressed metabolites; green: differentially expressed miRNA. Red box highlights *DCK*, one of the nine genes identified as a classifier of GC response (see *Figure 4C*). (**C**) Module decomposition of the hypernetwork. Genes modules (hexagons representing multiple highly connected genes) named by the most central gene in each module. miR-122-5p is present in the core of two modules (shown); color of modules represents centrality hierarchy: red: most central in the network; green: least central in the network.

*Figure 5 continued on next page*

Figure 5 continued

The online version of this article includes the following figure supplement(s) for figure 5:

**Figure supplement 1.** Heat map with clusters of circulating 'omic data associated with glucocorticoid exposure identified using a correlation matrix.

## Discussion

In a clinical experimental study designed to identify biomarkers of GC action, we succeeded in generating two profoundly different states of GC exposure within the physiological range in the same individual. The novelty of this study is the identification of pathways related to GC response and putative biomarkers of GC action in gene expression, metabolome, and miRNAs derived from integrated multi-omic analysis in two independent tissues. We identified a transcriptomic profile that was under similar GC regulation in both PBMC and adipose tissue transcriptomes, which was then validated by comparison to a range of previously published data by other research groups from cellular assays. We also identified a circulating miRNA, miR-122-5p, which was correlated with the circulating transcriptome and metabolome findings, suggesting for the first time a functional role in GC action. Moreover, the association between the expression of miR-122-5p and GC exposure was replicated in three independent study groups.

In order to identify putative biomarkers of GC action in humans, a clinical study was considered to be the most appropriate experimental setting. Addison's disease or primary adrenal insufficiency is a rare disorder, but a unique clinical model for GC biomarker discovery due to absent or very low endogenous GC production (*Gan et al., 2014*; *Sævik et al., 2020*). Subjects with Addison's disease were studied in a random order during physiological GC exposure and GC withdrawal. During GC exposure, infusion of HC delivered in isotonic saline via an infusion pump using a circadian pattern and saline alone (using the same volume and infusion pattern as during HC infusion) was

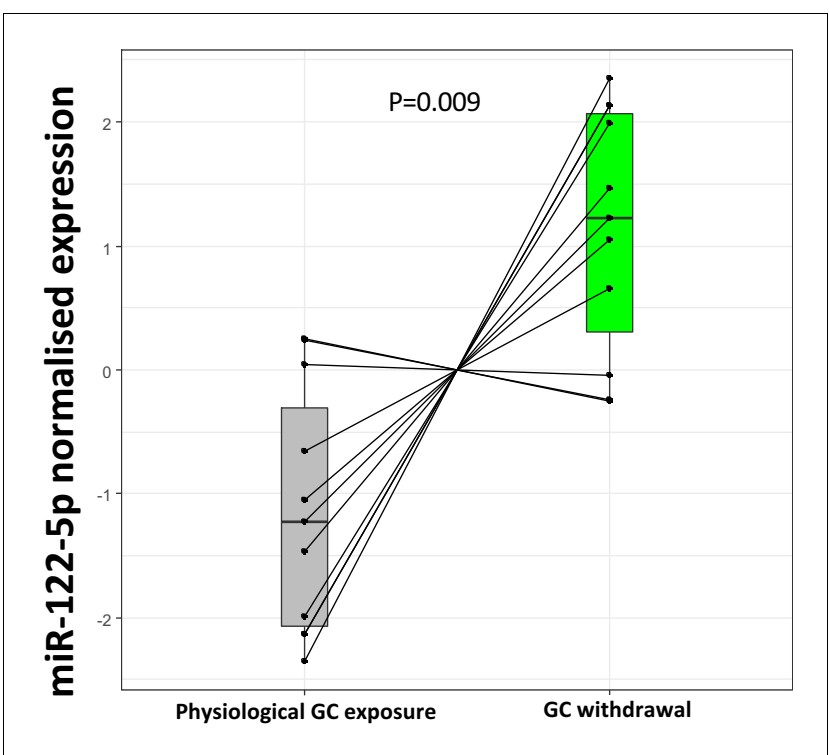

**Figure 6.** Replication of miR-122-5p as a putative biomarker of glucocorticoid (GC) action in the current biomarker discovery study. Targeted replication of plasma miR-122-5p fold change from the current study population between subjects with Addison's disease during GC exposure and GC withdrawal showed a significant down-regulation of miR-122-5p expression with increased GC exposure (p=0.009) conducted using an independent RNA separation procedure in the same samples.

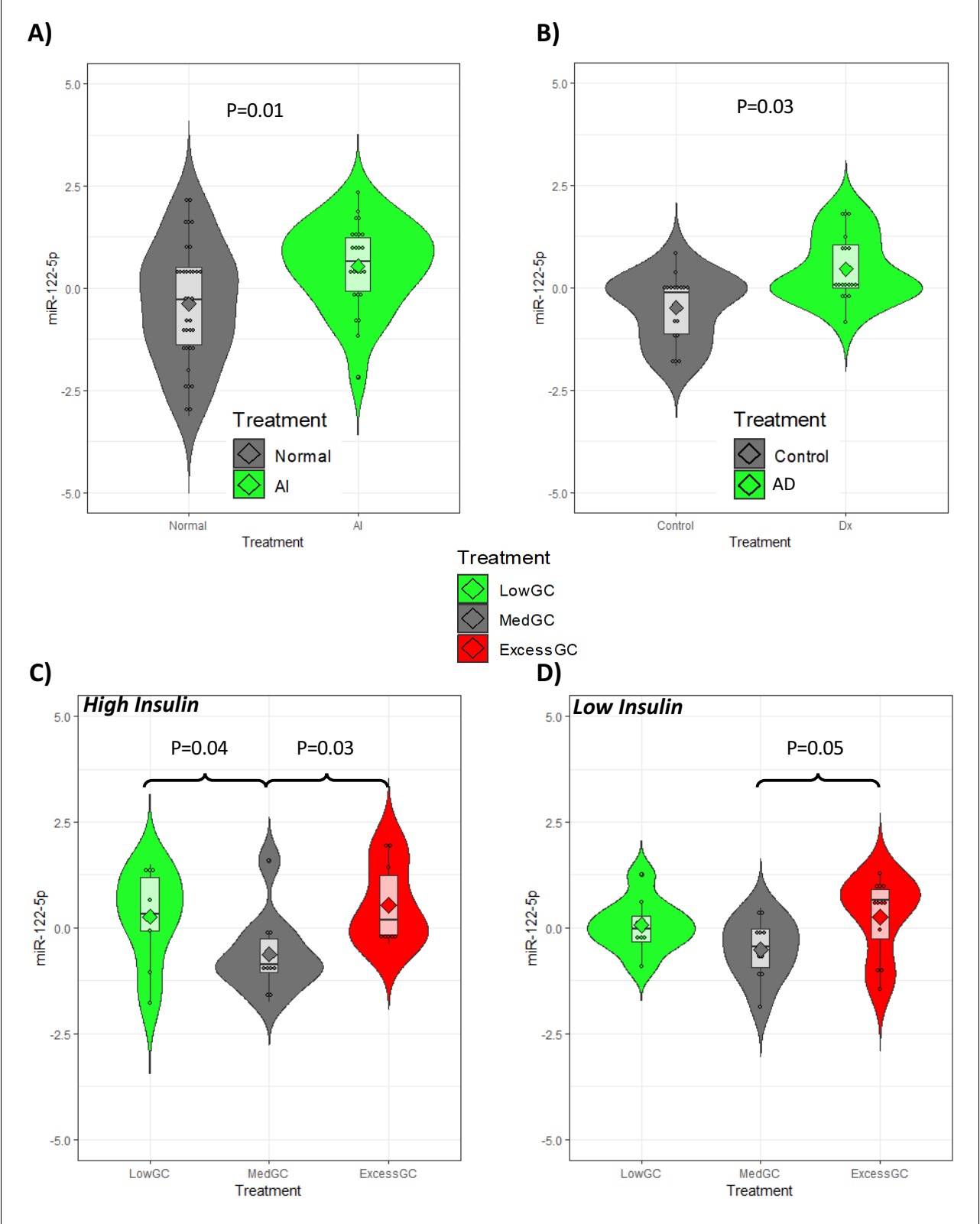

**Figure 7.** Replication of miR-122-5p as a putative biomarker of glucocorticoid (GC) action in independent patient groups with different GC exposure. (**A**) The expression of miR-122-5p was higher in subjects with rheumatoid arthritis and reduced GC exposure due to tertiary adrenal insufficiency after a short-term stop of the GC treatment (AI) than in those without tertiary adrenal insufficiency (Normal). (**B**) Subjects with Addison's disease (AD) had higher expression of miR-122-5p than healthy matched controls (Control). In an experimental study in healthy subjects, the expression of miR-122-5p

*Figure 7 continued on next page*

*Figure 7 continued*

was increased both after low and excessive high GC exposure (LowGC and ExcessGC, respectively) compared to medium GC exposure (MedGC) at both (**C**) high and (**D**) low serum insulin levels. Diamond: mean. Box = median ± interquartile range. Whiskers = upper and lower quartiles. miR-122-5p axis is presented as normalized expression.

administered during the GC withdrawal in order to prevent a state of sodium and fluid deficiency. This study design therefore allowed a within-individual comparison accounting for circadian rhythm and food intake. The marked difference in serum and urinary cortisol and cortisone, and the similar serum electrolytes, glucose, body weight, and blood pressure between the two interventions support the experimental success of the study design and strongly indicate that confounders related to metabolic changes or other secondary events related to the GC exposure or GC withdrawal were not influencing the output of the study. The measurable but very low concentrations of serum cortisol and cortisone throughout the GC withdrawal may be explained by a residual adrenal steroid secretion in some subjects (*Gan et al., 2014*; *Sævik et al., 2020*) and/or due to conversion of cortisone to cortisol in the liver and adipose tissue (*Stimson et al., 2014*).

Network models of 'omic data can be used as a framework to assess the potential utility of biomarkers (*Stevens et al., 2014*). In this study, we have used a hypernetwork model of GC action based on differential gene expression in PBMCs as a basis to integrate adipose tissue transcriptome, plasma miRNA, and serum metabolomic data. Hypernetwork analysis leverages the power inherent in large data sets to assess interactions between 'omic elements in a manner that is robust to false positives (*Battiston et al., 2020*). The associated interactome network derived from the PBMC transcriptome was shown to contain a number of genes with previously known GC-dependent binding of *NR3C1* (the GC receptor) to regulatory elements, evidence that supports the specificity of the study design (*Davis et al., 2018*; *Casper et al., 2018*). Gene ontology analysis of the differential gene expression identified a range of pathways classically associated with GC action including GC-receptor signaling, immunoregulatory pathways such as those involving NF-κB, metabolic pathways, and cell cycle pathways. The plasma miRNA and serum metabolomic data was shown to map to the interactome network model of GC action, and this was taken as support for this data being putative circulating biomarkers functionally related to GC action.

Differential expression induced by GC treatment in both PBMCs and adipose tissue was indirectly associated with similar downstream elements by gene ontology analysis. These genes were not directly implicated with GC response, so, while the exact mechanisms may be different in each tissue, effects are coordinated through the same elements. Integration of the multi-omic data including both PBMC and adipose tissue transcriptomes was performed in order to increase the robustness of putative markers that could reflect action in other tissues such as adipose tissue, which is an important target organ for the metabolic actions of GCs. The 59 genes that behaved similarly in PBMC and adipose tissue were then validated in a range of studies examining GC response in different cellular systems. These included primary cell culture on keratinocytes (*Stojadinovic et al., 2007*) and lens epithelial cells (*Gupta et al., 2005*), along with PBMCs (*Carlet et al., 2010*) and cancer cells [both lymphoblastic leukemia (*Carlet et al., 2010*) and osteosarcoma (*Lu et al., 2007*; *Jewell et al., 2012*)]. The set of nine genes co-regulated in relation to GC exposure and GC withdrawal in both PBMC and adipose tissue transcriptomes can therefore be considered as putative markers of GC response. These could be used as a gene set to interrogate GC action in other experimental settings.

All the miRNA findings in this study are novel. While emerging experimental evidence indicates impact on regulation of GC action at several points by miRNAs (*Clayton et al., 2018*), this is the first time that miRNAs are shown to be globally correlated to GC action in humans. Both the hypernetwork analysis and the interactome network model implied the functional significance of some miRNAs, particularly miR-122-5p. In our hypernetwork model, the expression of miR-122-5p was correlated with clusters of genes that were centrally coordinated by expression of both *RNF157* and *TBXAS1*, the former suggested to be a key regulator of both *PI3K* and *MAPK* signaling pathways, commonly perturbated in cancer and metabolic disorders (*Dogan et al., 2017*). Expression of TBXAS1 is pharmacogenomic linked to inhaled GC exposure in asthma (*Dahlin et al., 2020*). miR-122 is precursor transcript of mature miRNAs, including miR-122-5p (*Carthew and Sontheimer, 2009*; *Bartel, 2004*). miR-122 is expressed in the liver in humans (*Tsai et al., 2009*;

*GTEx Consortium, 2015*; *GTEx Consortium, 2013*) and mice (*Tsai et al., 2009*). Hepatocyte nuclear factor HNF4A (*Li et al., 2011*; *Xu et al., 2010*), along with HNF3A (FOXOA1), HNF3B (FOXOA2), and HNF1A (*Xu et al., 2010*; *Coulouarn et al., 2009*), has been shown to be a key regulator of miR-122 expression in human cells. Down-regulation of miR-122 in murine models has been associated with non-alcoholic fatty liver disease (*Alisi et al., 2011*) and diabetes mellitus (*Guay et al., 2011*), and in humans, miR-122-5p has also been associated with fatty liver disease (*Raitoharju et al., 2016*).

miR-122-5p may be a functional link between unphysiological GC exposure and metabolic and cardiovascular disease. Increased exposure to GCs impairs glucose tolerance and may induce type 2 diabetes (*Hackett et al., 2014*). Indeed, reduced miR-122-5p expression has been seen in animal models of diabetes, and the reduction of this miRNA in response to increased GC exposure may suggest that miR-122-5p is a functional link between GC action and metabolism. In support of these findings are observations showing that miR-122-5p regulate insulin sensitivity in murine hepatic cells by targeting the insulin-like growth factor (IGF) 1 receptor (*Dong et al., 2019*). Recent human studies have also suggested that miR-122-5p is an indicator of the metabolic syndrome, with reduced expression in response to weight loss in overweight/obese subjects (*Hess et al., 2020*). miR-122-5p has also been suggested as a biomarker of coronary artery stenosis and plaque instability (*Wang et al., 2019*; *Singh et al., 2020*; *Ling et al., 2020*). As unphysiological GC exposure has been associated with obesity, diabetes, and cardiovascular disease (*Walker, 2007*), it is possible that miR-122-5p is reflecting different GC exposure in these disorders. The subjects with Addison's disease in our clinical experimental study had no other comorbidities previously known to be associated with miR-122-5p expression, and therefore the presence of such confounders in our miR-122-5p finding seems to be unlike.

Specific miRNAs circulating in a stable, cell-free form in plasma or serum may serve as biomarkers in some diseases (*Kroh et al., 2010*), and, in our integrated analysis, they seem to be a realistic and clinically useful marker of GC action. We therefore focused on the replication of the miRNA findings from the discovery study. For this purpose, we performed a targeted analysis of 12 putative miRNAs and analyzed them in 120 subjects from independent study groups with different GC exposure in terms of dose, duration of exposure, and route of administration. The rationale for selecting these groups was that their GC exposure mostly remained within the normal physiological range. Despite the experimental differences between these studies, and the fact that these studies were not designed to study miRNA biomarkers of GC action, miR-122-5p was down-regulated by increased GC exposure in all of them. One exception was when short-term excessively high GC exposure was studied in afternoon samples in 20 subjects. There is no clear explanation for this, except the possibility that high non-physiological GC exposure has other secondary effects that may affect the levels of miR-122-5p.

The network analysis also identified putative metabolomic markers of GC action. GCs have a key role in metabolic regulation of stress by mobilizing energy through glucose, protein, and lipid metabolism. Previous studies have found an association between different GC doses and levels of branched-chain amino acids, fatty acids, some acyl carnitines, and tryptophan and its metabolites (*Alwashih et al., 2017a*; *Sorgdrager et al., 2018*). In our study, the amino acid tyrosine and the pyrimidine base uracil had a central position in the hypernetwork, which defined a group of highly connected multi-omic relationships within physiological GC exposure. Some of the other metabolomic data from our study was also in line with previous metabolomic studies in patients with adrenal insufficiency (*Alwashih et al., 2017b*; *Sorgdrager et al., 2018*). Excessive exposure to GCs in healthy subjects has, on the other hand, shown a strong, immediately and long-lasting impact on numerous biological pathways in the metabolome that may be either direct or indirect through the metabolic and cardiovascular action of pharmacological doses of GCs (*Bordag et al., 2015*).

There are some study limitations that need to be acknowledged. The low number of subjects included in the clinical experimental study could have reduced the power to detect a putative marker in individual 'omic data sets, but this limitation was compensated for by the crossover study design and the integration of multi-omic layers. Another limitation is that we have only studied markers collected in the morning during physiologically peak cortisol exposure. However, the strengths of our study are the experimental study design, consideration of diurnal variation in GC action and impact of food intake, and the within-individual comparison, which minimizes confounders, as well as the fact that the putative markers that we have replicated are associated with known

GC-responsive genes in two different tissues, suggesting their functional importance in GC action. Moreover, the integration of multi-omic layers allows for the reduction of background noise (*Huang et al., 2017*) and forms the basis for a detailed model of GC action. Hypernetwork summaries of correlation networks are recognized as providing signatures of mechanism (*Pearcy et al., 2016*; *Johnson, 2011*; *Butte et al., 2000*; *Oldham et al., 2006*) and, as such, are useful to assess both function and define markers of direct action.

In this clinical biomarker discovery study, we identified genes, miRNA, and metabolites that are differently expressed during GC exposure and GC withdrawal in subjects with Addison's disease. The multi-omic data showed a high degree of coherence, and network analysis identified transcriptomics and metabolites that were closely correlated. The final outcome of the study is identification of a miRNA that is regulated by GC exposure and correlated with genes and metabolites that are also regulated by GCs in this study, indicating its functional relevance. The replication of this miRNA in three independent study groups increases the likelihood that the discovered miRNA, miR-122-5p, could become a biomarker of GC action to be used in clinical settings.

# Materials and methods

## Key resources table

| Reagent type (species) or resource | Designation | Source or reference | Identifiers | Additional information |
|---|---|---|---|---|
| Recombinant DNA reagent | GeneChip WT PLUS Reagent Kit | Affymetrix Inc | Other | P/N 703174 Rev. 1 |
| Commercial assay or kit | Human Gene 1.0 ST array | Affymetrix Inc | – | – |
| Commercial assay or kit | Human Gene 1.1 ST array | Affymetrix Inc | – | – |
| Commercial assay or kit | Human Gene 2.0 ST array | Affymetrix Inc | – | – |
| Commercial assay or kit | miRCURY LNA Universal RT microRNA PCR, Polyadenylation, and cDNA Synthesis Kit | Exiqon | – | – |
| Commercial assay or kit | miRCURY RNA Isolation Kit-Biofluids | Exiqon | – | – |
| Chemical compound, drug | Solu-Cortef | Pfizer Inc | – | – |
| Software, algorithm | Agilent Masshunter Profinder | Agilent Technologies, Inc | Other | Version B.08.00 |
| Software, algorithm | SPSS | SPSS | RRID:SCR_002865 | – |
| Software, algorithm | R | R Project for Statistical Computing | RRID:SCR_001905 | – |
| Software, algorithm | Rstudio | Rstudio | RRID:SCR_000432 | – |
| Software, algorithm | MetaboAnalystR | – | RRID:SCR_016723 | – |
| Software, algorithm | Moduland algorithm | – | https://www.linkgroup.hu/docs/ModuLand-ESM1-v5.pdf | – |
| Software, algorithm | Cytoscape | Cytoscape | RRID:SCR_003032 | – |
| Software, algorithm | Qlucore | Qlucore Omics Explorer | https://www.qlucore.com/bioinformatics | – |
| Software, algorithm | Robust Multi-Array Average algorithm | – | http://www.molmine.com | – |
| Software, algorithm | ChromaTOF | LECO | https://www.leco.com | – |
| Software, algorithm | MATLAB R2016a | Mathworks | https://www.mathworks.com | – |
| Software, algorithm | Roche LC software | Roche Molecular Systems, Inc | – | – |
| Software, algorithm | NormFinder | Aarhus University Hospital, Denmark | RRID:SCR_003387 | – |

*Continued on next page*

*Continued*

| Reagent type (species) or resource | Designation | Source or reference | Identifiers | Additional information |
|---|---|---|---|---|
| Software, algorithm | NIST MS 2.0 software | NIST | https://chemdata.nist.gov | – |
| Other | LightCycler 480 Real-Time PCR System | Roche Molecular Systems, Inc | RRID:SCR_020502 | – |
| Other | Agilent 1290 Infinity UHPLC-system | Agilent Technologies, Inc | https://www.agilent.com | – |
| Other | Agilent 2100 Bioanalyzer system | Agilent Technologies, Inc | RRID:SCR_018043 | – |
| Other | Agilent 6550 iFunnel Q-TOF LC/MS | Agilent Technologies, Inc | RRID:SCR_019433 | – |
| Other | ENCODE | Stanford University | RRID:SCR_015482 | – |
| Other | UCSC Genome Browser | UCSC | RRID:SCR_005780 | – |
| Other | TarBase | DIANA Tools | RRID:SCR_010841 | – |
| Other | miRecords | Biolead.org | RRID:SCR_013021 | – |
| Other | TargetScan | Whitehead Institute for Biomedical Research | RRID:SCR_010845 | – |
| Other | BioGRID | TyersLab.com | RRID:SCR_007393 | – |
| Other | Ingenuity Pathway Analysis | Qiagen | RRID:SCR_008653 | – |

## Experimental study design

### Study design
The study was a prospective, single-center, single-blind, randomized, two-period/crossover clinical trial.

### Study subjects
Men and women with Addison's disease for >12 months on stable cortisol replacement (with HC 15–30 mg/day) for $\geq$3 months followed at the Center for Adrenal diseases in the Out-patient Clinic at the Department of Endocrinology-Diabetes-Metabolism, Sahlgrenska University Hospital (tertiary referral hospital), Gothenburg, Sweden, were eligible for inclusion. Other inclusion criteria were age 20–60 years, body mass index 20–30 kg/m$^2$, and ability to comply with the protocol procedures. Exclusion criteria were GC replacement therapy for indication other than Addison's disease, any treatment with sex hormones including contraceptive drugs, treatment with levothyroxine, renal or hepatic failure, significant and symptomatic cardiovascular disease, diabetes mellitus, current infectious disease with fever, and pregnancy or breastfeeding. Recruitment was stopped when all eligible subjects had been asked to participate.

Power calculation was not performed because of the exploratory nature of the study. Power calculations were also difficult in the context of 'omic analysis as there may be variable effect sizes over different 'omic elements.

The study was approved by the Ethics Review Board of the University of Gothenburg, Sweden (permit no. 374-13, 8 August 2013) and conducted in accordance with the Declaration of Helsinki. Written informed consent was obtained from all subjects before participation. The study was registered at ClinicalTrials.gov with identifier NCT02152553.

### Study treatment
HC infusion was prepared by adding 0.4 mL of Solu-Cortef 50 mg/mL to 999.6 mL 0.9% saline, which resulted in 1 mg HC per 50 mL intravenous infusion. HC infusion was adjusted in accordance with previous observations in healthy males (*Kerrigan et al., 1993*) and interventions in both sexes (*Løvås and Husebye, 2007*; *Figure 1B*). The aim was to achieve a near-physiological circadian cortisol curve with early morning rise in serum cortisol that would peak at 7 AM and trough concentrations at midnight. In the GC-withdrawal intervention, 0.9% saline infusion alone was administered

using the same volume as during the HC infusion. Thus, a person weighing 75 kg received 2 L of intravenous infusion over 22 hr during each intervention.

## Interventions

All subjects were admitted after an overnight fast to the in-patient Endocrinology Department at the Sahlgrenska University Hospital at 8 AM (first intervention day) and were discharged at 12 PM the following day (second day). Subjects were randomized using a free randomization plan (generated at http://www.randomization.com/ on 27 April 2014) before the first intervention to receive either HC infusion or only saline infusion in a single-blind, crossover manner at least 2 weeks apart (*Figure 1B*). The researcher responsible for the clinical study generated the randomization plan, enrolled the study subjects, and assigned participants to interventions. Female subjects (when fertile) were studied during the early follicular phase (days 5–10) of their regular cycle under both interventions. Subjects were told not to take their ordinary mineralocorticoid dose on the day before each intervention but to take their ordinary HC dose. Subjects received standard meals at fixed times during both interventions. Their consumption of coffee or tea was recorded during the first intervention in order to consume the same amount and at the same time points during the second intervention.

During each intervention, the subjects' blood pressure, body temperature, and weight were monitored. Because of the study design and the variations in circadian rhythm, blood sampling was collected at exactly the same time before the start of intervention, at midnight (12 AM), and in the morning of the second intervention day (7 AM). Urine was collected between midnight and morning (overnight), and abdominal subcutaneous fat was collected in the morning of the second intervention day immediately after blood and urine sampling. Adipose tissue was collected after local injection with lidocaine under the umbilicus on the right side of the abdomen during saline infusion and on the left side during HC infusion. The study was unblinded for each study subject after the completion of all aspects of the study (the second intervention).

## Replication studies

### Baseline samples in subjects treated with prednisolone for rheumatoid arthritis

This was a cross-sectional clinical study of prednisolone-induced adrenal insufficiency undertaken at the Department of Medical Endocrinology and Metabolism, at University Hospital, Rigshospitalet, Copenhagen, Denmark, between 2012 and 2018 (*Borresen et al., 2017*). In the current replication analysis, 60 subjects were included. All subjects had rheumatoid arthritis, received long-term prednisolone treatment (minimum 6 months), and treated with a current prednisolone dose of 5 mg/day. Of the 60 subjects, 23 had an insufficient response to the Synacthen test (GC-induced adrenal insufficiency, AI group) and 37 had a normal response (normal group). The samples included in the replication analysis were collected in the morning after an approximately 48 hr pause of prednisolone dosing (before the Synacthen test) and after overnight fasting. Plasma miRNA analysis of frozen samples was performed at Exiqon Services, Denmark.

### Case–control study in subjects with or without Addison's disease

This was an observational, cross-sectional, single-center, case–control study undertaken in our unit in Gothenburg, Sweden, between 2005 and 2009 (*Bergthorsdottir et al., 2017*). In the current replication analysis, the subgroup of 20 subjects with Addison's disease under daily replacement therapy with oral HC $\geq$ 30 mg (AD group) and their 20 healthy control subjects with no GC therapy matched for age and gender (control group) were included. The samples included in the replication analysis were collected in the morning between 8 AM and 10 AM after an overnight fast, and for the cases after morning administration of their oral HC, which means a very low cortisol exposure during the night before sample collection. Plasma miRNA analysis of frozen samples was performed at Exiqon Services, Denmark.

### Randomized, crossover study in healthy subjects

This was a randomized, double-blind study in 20 lean healthy male volunteers undertaken at the Edinburgh Clinical Research Facility between July 2010 and April 2012. The full protocol has been published previously (*Stimson et al., 2017*). Volunteers were randomized to receive either a low- or

medium-dose insulin infusion (10 subjects in each group) and attended on three occasions after overnight fasting. Subjects received metyrapone (to inhibit adrenal cortisol secretion) with and without HC infusion (over 6.5 hr) in order to produce low, medium, or excessive GC levels (Low/Med/ExcessGC during high insulin and low insulin cohorts, respectively). The samples included in the replication analysis were collected in the afternoon at the end of each intervention (approximately 6.5 hr after start) on three occasions (low, moderate, or excessive high GC levels). Plasma miRNA analysis of frozen samples was performed at Exiqon Services, Denmark.

## Generation and preparation of 'omic data

Plasma cortisol and cortisone were analyzed using liquid chromatography-mass spectrometry (LC-MS), and urinary-free cortisol and cortisone were analyzed using gas chromatography-mass spectrometry (GC-MS) at the Mass Spectrometry Core Laboratory, Centre for Cardiovascular Science, University of Edinburgh, Edinburgh, UK. PBMCs were isolated on-site from whole blood using a gradient-based separation procedure and Ficoll-Paque PREMIUM (GE Healthcare).

A microarray gene expression analysis using Affymetrix Human Gene 2.0 ST arrays in both PBMC and adipose tissue was performed at the Array and Analysis Facility, Science for Life Laboratory at Uppsala Biomedical Center (BMC), Sweden.

The untargeted miRNA analysis in plasma was performed at Exiqon Services, Denmark. The targeted miRNA analyses in plasma (including the replication samples) were performed at Exiqon Services, Denmark, at a later date than the untargeted analysis. The 14 miRNAs included in the analysis based on the findings from the untargeted analysis were miR-425-3p, miR-186-5p, miR-15b-5p, miR-95-3p, miR-16-1-3p, miR-576-5p, miR-122-5p, miR-200a-3p, miR-193b-3p, miR-424-5p, miR-574-3p, miR-148a-3p, miR-18a-5p, and let-7g-5p.

Metabolic profiling of serum by GC-MS and LC-MS was performed at the Swedish Metabolomics Center in Umeå, Sweden.

Preprocessing of 'omics data sets was carried out in the following ways. PBMC and adipose tissue transcriptomes were normalized using robust multichip average (RMA) via the R package oligo (*Carvalho and Irizarry, 2010*), which corrects for background variation, quantile normalizes, and summarizes features to gene-probe set level (*Figure 3—figure supplement 2*). GC-MS and LC-MS metabolomic data sets were analyzed using the R package MetaboanalystR (*Chong and Xia, 2018*), which filters variables based on ranked interquartile range, normalizes metabolites to sample median, and log transforms the resultant intensities (*Figure 3—figure supplement 3*). Qlucore Omics Explorer (version 3.3, Lund, Sweden) was used to scale and mean center miRome data. How all these analyses were performed is described in detail in Appendix 3.

## Data analysis of differential gene expression

Principal component analysis (PCA) was performed to provide further quality control and define the relationship of variance between samples, allowing structure within the data set to be defined (Qlucore Omics Explorer 3.3). Quality control of transcriptomic data was performed using PCA with cross-validation and data consistency was confirmed. No outliers were identified. Differential gene expression was determined by a paired *t*-test comparing the two interventions. Network analysis of DEGs was performed using Advaita Bio's iPathwayGuide (https://www.advaitabio.com/ipathway-guide); gene ontology performed using this software analysis tool implements the 'Impact Analysis' approach that takes into consideration the direction and type of all signals on a pathway, and the position, role, and type of every gene (*Ahsan and Drăghici, 2017*).

## Gene ontology, gene expression regulated by miRNA, and causal network analysis

Gene ontologies were associated with differentially regulated gene lists (Ingenuity Pathway Analysis [IPA], Qiagen, Redwood City, CA). miRNAs were paired with genes that were theoretically regulated by specific miRNAs using IPA. The databases used for this mapping were TarBase (*Vlachos et al., 2015*), miRecords (*Xiao et al., 2009*), and peer-reviewed biomedical literature, as well as predicted miR–mRNA interactions from TargetScan (*Agarwal et al., 2015*).

The Encyclopedia of DNA Elements (ENCODE) data (*Rosenbloom et al., 2013*) was used to map genes in the interactome network model of GC action that had been previously shown to have dexamethasone dose-dependent DNA binding of *NR3C1*, the GC receptor gene.

Causal network analysis (CNA) allows the identification and prioritization of regulatory system elements within transcriptomic models. CNA was performed within IPA (*Krämer et al., 2014*). CNA identifies upstream molecules, up to three steps distant, that potentially control the expression of the genes in the data set (*Krämer et al., 2014*). A prediction of the activation state for each regulatory factor (master regulator), based on the direction of change, was calculated (Z-score) using the gene expression patterns of the transcription factor and its downstream genes. An absolute Z-score of $\geq |1.4|$ and a corrected p-value<0.05 (Fisher's exact test) were used to compare the regulators identified.

## Network model construction and comparison

Lists of DEGs were used to generate network models of protein interactions in Cytoscape 2.8.3 (*Smoot et al., 2011*) by inference using the BioGRID (3.4.137) database (*Chatr-Aryamontri et al., 2015*).

The Cytoscape plug-in Moduland (*Kovács et al., 2010*; *Szalay-Beko et al., 2012*) was applied to identify overlapping modules, an approach that models complex modular architecture within the human interactome (*Chang et al., 2013*) by accounting for the non-discrete nature of network modules (*Kovács et al., 2010*). Modular hierarchy was determined using a centrality score and further assessed using hierarchical network layouts (summarizing the underlying network topology). The central module cores (metanode of the 10 most central elements) was determined and used as a basis to integrate the miRNA and metabolomic data. Transcriptomic and metabolomic data were combined to form a single network model using the Metscape (*Karnovsky et al., 2012*) plug-in for Cytoscape. Differential 'omic data was compared and clustered in a correlation matrix using the corrplot plug-in (*Murdoch and Chow, 1996*) for R (*R Development Core Team, 2020*).

## Similarity network fusion

Subject-level similarity network fusion (SNF) (*Wang et al., 2014*) was performed on 'omic data as a test for similarity. To perform SNF, the *SNFTool* R-package was used (*Wang et al., 2014*). First, Euclidean distances were calculated between gene probe sets, and these were then combined using a nonlinear nearest neighbor method over 20 iterations. The fused data was subjected to spectral clustering and presented as a heat map.

## Hypernetworks

We modeled the dynamics of potentially relevant PBMC and adipose tissue transcripts, miRNAs, and metabolites by assessing their activity as measured by the number of shared correlations against the background of all 'omic elements called present after data processing.

A matrix (*m* rows and *n* columns) was generated of correlation distances (*r*-values) between the significantly differentially expressed multi-omic data (forming *m* rows) and all 'omic data called present (forming *n* columns). The *r*-values were normally distributed.

A similarity matrix was defined by dichotomizing the correlation distance based on an *r*-value threshold of $\geq |1.5|$sd (if sd of $|r| \geq 1.5$ , then value = 1; if sd of $|r|<1.5$, then value = 0); the new matrix was termed *M* and represents the incidence matrix of the hypernetwork. An element of *M*, $m_{ij}$, where *i* and *j* are elements of *m* and *n* respectively, is defined as follows:

$$m_{ij} = \begin{cases} 1, & \&|r| \geq 1.5 \; sd \\ 0, & \&|r|<1.5 \; sd \end{cases}$$

To generate the hypernetwork, we multiplied *M* by the transpose of *M*, $M^T$ (*Johnson, 2011*; *Ha et al., 2020*), the elements of the resulting square matrix (*sM*, an $m \times m$ matrix) are the number of correlations shared by each pair of interacting 'omic elements; this is also the number of edges connecting each pair of nodes. *sM* was clustered using hierarchical clustering to identify the group of highly connected 'omic elements.

The dichotomization parameters were shown to correspond to maximum signal window in the data using chi-squared distance metric (*Figure 3—figure supplement 4*). The chi-squared distance ($X^2$) was defined as

$$X^2 = \sum_{i=1}^{N} \frac{(m_i - m_e)^2}{m_e}$$

where $N$ is the order of the matrix $sM$, $i$ is the $i'th$ element, $m_i$, of $sM$, and $m_e$ is the expected value of an element of $sM$. The expected value of an element of $sM$ was calculated at any chosen dichotomization threshold by dividing the total number of correlations by the order of the matrix.

Differential expression analysis was performed to refine genes for hypernetwork analysis. This approach serves to identify potentially relevant 'omic elements. FDR-corrected p-values for all elements selected for hypernetwork integration are presented in *Supplementary file 1g*. We identified 4426 DEGs in PBMCs, 3520 adipose tissue DEGs, 38 metabolites (17 LC-MS, 21 GC-MS), and 12 miRNAs below an uncorrected p-value of 0.05. Data was analyzed across nine matching samples (normalized log2 score was inverted between GC exposure and GC withdrawal, i.e., +1 and –1, respectively).

A hypernetwork is inherently robust as individual correlations are not considered significant; rather hypernetworks model higher order interactions between nodes ('omic elements) based on large numbers of shared edges (correlations). This approach only highlights 'omic elements that are supported by the majority of the data and, as such, is robust to a wide range of *r*-value thresholds as well as small sample sizes.

Further, robustness of the hypernetwork observations was determined using a dissimilarity matrix derived from the original similarity matrix (i.e., the complement of the similarity matrix). The elements assessed as dissimilar were subtracted from those defined as similar. Elements within the $M \times M^T$ output of the dissimilarity analysis that were also similar were eliminated from further predictive analysis.

The BORUTA R package (*Kursa, 2014*; *Kursa and Rudnicki, 2010*) was used for feature selection of transcriptomic data with predictive value. Random Forest (*Breiman, 2001*) was implemented in R using 5000 trees to determine the predictive value expressed as the area under the curve of the receiver operating characteristic.

## Statistical analyses

Unsupervised analysis of metabolomic and transcriptomic data to assess how GC exposure grouped the study subjects was performed using Orthogonal Projections to Latent Structures Discriminant Analysis in SIMCA 13.0 (Sartorius) or PLS-DA MixOmics plug-in (*Rohart et al., 2017*) for R.

For quantitative variables with normal distribution, we performed independent samples *t*-test. Mann–Whitney *U*-test was performed for non-normally distributed variables. Chi-squared test or Fisher's exact test, as appropriate, was used for categorical variables. Wilcoxon rank test was used for detecting differences between the two interventions in quantitative non-normally distributed variables. All statistical tests were two-sided, and p<0.05 was considered to be statistically significant. Further robustness for 'omic data analysis was provided by considering the findings as clusters of co-expressed findings (*Cleary et al., 2017*). Statistical analyses were performed using SPSS (Statistical Package for Social Science) program, version 24 software for Mac.

Unless otherwise stated, all other statistical analyses were performed in R version 4.0.2 for Windows. Figures were plotted using ggplot2 (*Wickham, 2016*), gplots (*Warnes et al., 2020*), ggpubr (*Kassambara, 2020*), and reshape2 (*Wickham, 2007*).

## Acknowledgements

We are grateful to all of the study volunteers for their participation in the respective studies. We would also like to thank the following for their contribution to the project: the In-patient and Out-patient Clinics at the Department of Endocrinology-Diabetes-Metabolism, Sahlgrenska University Hospital, Gothenburg, Sweden, and especially the Center for Endocrinology and Metabolism; research nurse Lena Strindberg and nurses Olof Ehn, Ingrid Broms, Frida Gillberg, and Rebecka Starke; the Array and Analysis Facility, Science for Life Laboratory at Uppsala Biomedical Center

(BMC), Uppsala, Sweden; the Swedish Metabolomics Center in Umeå, Umeå, Sweden; Ruth Andrew and Natalie Homer at the Mass Spectrometry Core Laboratory, Clinical Research Facility, University of Edinburgh, Edinburgh, UK; and Peter Todd (Tajut Ltd., Kaiapoi, New Zealand) for third-party writing assistance in drafting this manuscript, for which he received financial compensation from ALF funding. The study was registered at ClinicalTrials.gov with identifier NCT02152553. The exploratory study and the analyses were supported by The Swedish Research Council (Project 2015-02561 and 2019-01112) and The Swedish federal government under the LUA/ALF agreement (Project ALFGBG-719531). DC was supported by The Swedish Endocrinology Association and The Gothenburg Medical Society. BW was supported by Wellcome Trust through an Investigator Award. RHS was supported by grants from The Medical Research Council (MR/K010271/1) and The Chief Scientist Office (SCAF/17/02). The rheumatoid arthritis study (replication study) was supported by The Eva Madura's Foundation, The Research Foundation of Copenhagen University Hospital, Rigshospitalet, and The Danish Rheumatism Association.

## Additional information

### Competing interests

Dimitrios Chantzichristos: DC has received lecture fees from Otsuka, Sanofi, and Shire. Brian R Walker: BW is a consultant with Actinogen Medical, inventor on patents owned by the University of Edinburgh relating to HSD1 inhibitors and to the discovery of glucocorticoid-sensitive biomarkers (the HSD1 inhibitor patents have been licensed to Actinogen Medical, patent numbers: WO 2011/135276, WO 2011/033255, and WO 2009/112845); BW is Non-Executive Director of Roslin Technologies Ltd. Oskar Ragnarsson: OR has received lecture fees from Novo Nordisk, Ipsen, Sandoz, and Pfizer, an unrestricted research grant from HRA Pharma, and consultancy fees from Novartis and HRA Pharma. Penelope Trimpou: PT has received lecture fees from Novartis and Novo-Nordisk. Adam Stevens: AS has received lecture fees from Merck. Gudmundur Johannsson: GJ has served as a consultant for Novo Nordisk, Shire, and Astra Zeneca and has received lecture fees from Eli Lilly, Ipsen, Novartis, Novo Nordisk, Merck Serono, Otsuka, and Pfizer. The other authors declare that no competing interests exist.

### Funding

| Funder | Grant reference number | Author |
| --- | --- | --- |
| Vetenskapsrådet | Project 2015-02561 and 2019-01112 | Gudmundur Johannsson |
| The Swedish federal government under the LUA/ALF agreement | Project ALFGBG-719531 | Gudmundur Johannsson |
| The Swedish Endocrinology Association | | Dimitrios Chantzichristos |
| Gothenburg Medical Society | | Dimitrios Chantzichristos |
| Wellcome Trust | Investigator Award | Brian R Walker |
| Medical Research Council | MR/K010271/1 | Roland H Stimson |
| Chief Scientist Office | SCAF/17/02 | Roland H Stimson |
| The Eva Madura's Foundation | | Ulla Feldt-Rasmussen |
| Rigshospitalet | | Ulla Feldt-Rasmussen |
| Danish Rheumatism Association | | Ulla Feldt-Rasmussen |

The funders had no role in study design, data collection and interpretation, or the decision to submit the work for publication.

## Author contributions
Dimitrios Chantzichristos, Conceptualization, Resources, Data curation, Formal analysis, Funding acquisition, Validation, Investigation, Visualization, Methodology, Writing - original draft, Project administration, Writing - review and editing; Per-Arne Svensson, Conceptualization, Resources, Supervision, Validation, Investigation, Methodology, Project administration, Writing - review and editing; Terence Garner, Data curation, Software, Formal analysis, Validation, Visualization, Methodology, Writing - review and editing; Camilla AM Glad, Per-Anders Jansson, Resources, Investigation, Methodology, Writing - review and editing; Brian R Walker, Conceptualization, Resources, Funding acquisition, Validation, Methodology, Project administration, Writing - review and editing; Ragnhildur Bergthorsdottir, Investigation, Methodology, Writing - review and editing; Oskar Ragnarsson, Penelope Trimpou, Resources, Methodology, Writing - review and editing; Roland H Stimson, Resources, Funding acquisition, Validation, Investigation, Methodology, Writing - review and editing; Stina W Borresen, Resources, Validation, Investigation, Methodology, Writing - review and editing; Ulla Feldt-Rasmussen, Resources, Supervision, Funding acquisition, Validation, Investigation, Methodology, Writing - review and editing; Stanko Skrtic, Conceptualization, Resources, Supervision, Validation, Methodology, Project administration, Writing - review and editing; Adam Stevens, Conceptualization, Data curation, Software, Formal analysis, Supervision, Validation, Visualization, Methodology, Writing - original draft, Writing - review and editing; Gudmundur Johannsson, Conceptualization, Resources, Supervision, Funding acquisition, Validation, Investigation, Visualization, Methodology, Writing - original draft, Project administration, Writing - review and editing

## Author ORCIDs
Dimitrios Chantzichristos (iD) https://orcid.org/0000-0002-1660-1973
Brian R Walker (iD) https://orcid.org/0000-0002-2416-1648
Roland H Stimson (iD) https://orcid.org/0000-0002-9002-6188
Adam Stevens (iD) https://orcid.org/0000-0001-8335-4143

## Ethics
Clinical trial registration: The study was registered at ClinicalTrials.gov with identifier NCT02152553. Human subjects: The study was approved by the Ethics Review Board of the University of Gothenburg, Sweden (permit no. 374-13, 8 August 2013) and conducted in accordance with the Declaration of Helsinki. Written informed consent was obtained from all subjects before participation.

## Decision letter and Author response
Decision letter https://doi.org/10.7554/eLife.62236.sa1
Author response https://doi.org/10.7554/eLife.62236.sa2

# Additional files
## Supplementary files
• Supplementary file 1. Analysis of 'omic datasets. (1) S file 1a. Transcriptomic analysis of peripheral blood mononuclear cells and adipose tissue from 7 AM (morning of the second intervention day) treated with hydrocortisone and saline (control). (2) S file 1b. miRNA with differential expression in plasma samples from 7 AM (morning of the second intervention day) treated with hydrocortisone and saline (control). (3) S file 1c. Gas chromatography-mass spectrometry (GC-MS) of serum samples from 7 AM (morning of the second intervention day) treated with hydrocortisone and saline (control). (4) S file 1d. Liquid chromatography-mass spectrometry (LC-MS) of serum samples from 7 AM (morning of the second intervention day) treated with hydrocortisone and saline (control). (5) S file 1e. The set of 59 genes with fold changes in the same directions in peripheral blood mononuclear cells (PBMCs) and adipose tissue transcriptomic data sets from 7 AM (morning of the second intervention day) treated with hydrocortisone and saline (control). (6) S file 1f. The order of the 11 clusters of 'omic data in the correlation matrix. (7) S file 1g. Analysis of all 'omic sets at 7 AM (morning of the second intervention day) treated with hydrocortisone and saline (control). (8) S file 1h. Biological pathways associated with differential gene expression in PBMCs between hydrocortisone and saline treated patients at 7 AM on the second intervention day. (9) S file 1i. The central gene in each

network module arranged in hierarchical order as ranked by network centrality score. (10) S file 1j. Causal network analysis of the differential gene expression in PBMCs associated with glucocorticoid action.

- Transparent reporting form
- Reporting standard 1. CONSORT 2010 Randomised Trial Checklist.
- Reporting standard 2. CONSORT 2010 Flow Diagram.

## Data availability

Transcriptomic data are available on the Gene Expression Omnibus (GEO) - GSE148642. Metabolomic and miRNAomic data are available through Mendeley Data - https://doi.org/10.17632/7hc49hzzhc.1.

The following datasets were generated:

| Author(s) | Year | Dataset title | Dataset URL | Database and Identifier |
|---|---|---|---|---|
| Stevens A | 2020 | Biocort | http://dx.doi.org/10.17632/7hc49hzzhc.1 | Mendeley Data, 10.17632/7hc49hzzhc.1 |
| Stevens A | 2021 | Biocort | https://www.ncbi.nlm.nih.gov/geo/query/acc.cgi?acc=GSE148642 | NCBI Gene Expression Omnibus, GSE148642 |

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

## Appendix 1

## Differentially regulated genes, miRNAs, and metabolites

A whole-genome transcriptomics analysis was performed separately from mRNA extracted from (a) PBMCs and (b) abdominal subcutaneous fat both collected at 7 AM. The RMA algorithm was used for background correction and quantile normalization (*Irizarry et al., 2003*). Gene-level summarization of exon-level data was conducted using the Affymetrix Human Gene 2.0 ST annotation. Processing was conducted using the R-packages 'oligo' and 'Limma'. Data homogeneity was assessed using PCA (Qlucore Omics Explorer and limma) with cross-validation. No outliers were identified. Differential gene expression was determined by linear models using the lmFit function in limma and by group analysis of variance (ANOVA). Both p-values for group ANOVA and FDR-modified p-values for linear models (Benjamini–Hochberg method [*Benjamini and Hochberg, 1995*]) were reported. By comparing GC exposure to GC withdrawal, 289 DEG probe sets were identified in PBMCs and 141 in adipose tissue (each FDR-modified $p<0.05$), consisting of 234 and 111 unique known genes, respectively (*Supplementary file 1a*). Five of the DEGs occurred in both PBMC and adipose tissue transcriptomes (*FOXO1, KLF9, PER1, TXNIP,* and *ZBTB16*), of which four had the same direction of expression (*Supplementary file 1a*).

In total, 9 out of 252 analyzed plasma miRNAs were differentially expressed at 7 AM between the two interventions ($p<0.05$, paired *t*-test and Wilcoxon signed-rank test) (*Supplementary file 1b*). By mapping to databases of predicted interactions between miRNA and gene transcripts (Ingenuity Pathway Analysis), these nine miRNAs were identified as possible regulators of 46 of the 234 unique DEGs from the PBMC transcriptome analysis (all known to be GC responsive) (*Supplementary file 1b*).

For all metabolomic data, Metaboanalyst (via R package MetaboanalystR) was used to filter variables based on ranked interquartile range, normalize metabolites to sample median, and log transform the resultant intensities. Comparison of 164 metabolite fragments (82 from GC-MS and 82 from LC-MS) between the two interventions at 7 AM revealed a distinction between GC exposure and GC withdrawal. A paired *t*-test analysis identified 21 metabolite fragments from the GC-MS and 17 metabolite fragments from the LC-MS analysis that were significantly different ($p<0.05$) (*Supplementary file 1c, d*).

The 'omic data sets from PBMCs, plasma miRNA, and serum metabolomics were examined for inter-subject variability. SNF was used to show that the integrated unsupervised 'omic data sets were fundamentally homogenous across the study subjects (*Appendix 1—figure 1A*). Some evidence of grouping between study subjects related to specific pair-wise comparison of 'omic data sets was observed (*Appendix 1—figure 1B*).

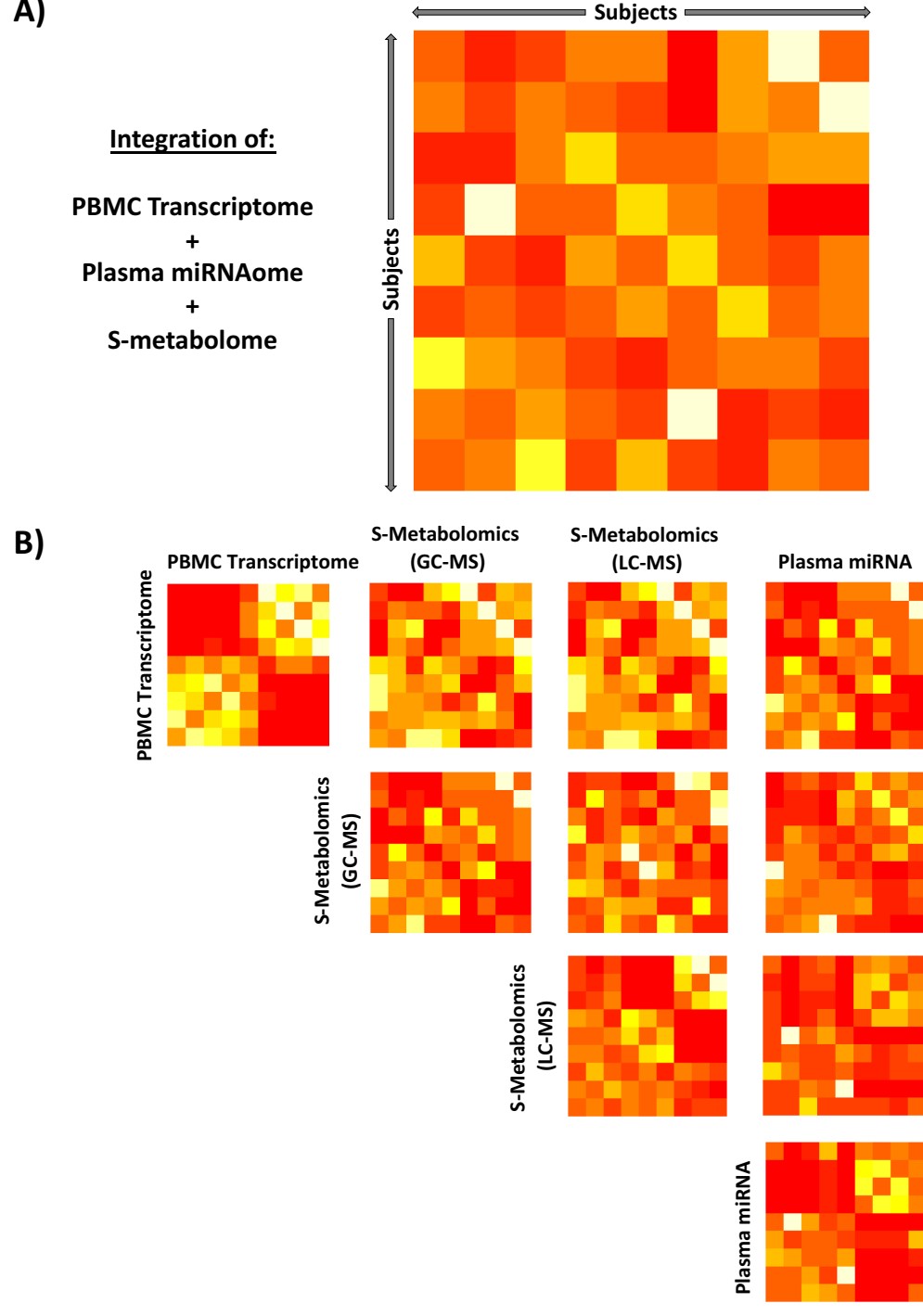

**Appendix 1—figure 1.** Integration of unsupervised peripheral blood mononuclear cell (PBMC) transcriptomic, metabolomic, and miRNAomic data in subjects between glucocorticoid (GC) exposure and GC withdrawal at 7 AM using similarity network fusion. Integration of all circulating 'omic data (**A**) simultaneously and (**B**) stepwise. Each matrix represents a subject-by-subject comparison using similarity network fusion of change in signal of 'omic data between GC exposure and GC withdrawal. The Euclidean distances between subjects based on their relative positions as determined by isomap dimensional scaling were used as a basis to determine similarity. Colors denote normalized continuous similarity score; red: dissimilar; yellow: similar. The complete data set for each specific 'omic layer of data was used, transcriptome n = 53,617, GC-MS n = 82, LC-MS

*Appendix 1—figure 1 continued on next page*

*Appendix 1—figure 1 continued*

$n$ = 82, miRNA $n$ = 251. GC-MS, gas chromatography-mass spectrometry; LC-MS: liquid chromatography-mass spectrometry; miRNA, microRNA; S, serum.

## Appendix 2

### Gene ontology and interactome network modelling

Biological pathways associated with the differential gene expression observed between the two interventions in PBMCs collected at 7 AM were assessed by gene ontology (*Appendix 2—figure 1A* and *Supplementary file 1h*). A range of classically GC-responsive pathways were over-represented during GC exposure (–log p-value range, 1.3–3.6), including GC receptor signaling and NF-κB signaling, along with a range of metabolism-linked pathways such as sphingosine-1-phosphate signaling, insulin growth factor-1 signaling, and 3-phosphoinositide biosynthesis.

We generated an interactome network model (with 2467 nodes [proteins/genes]) inferred from the 234 unique known genes with differential expression between GC exposure and GC withdrawal and all known protein:protein interactions (BioGRID database; *Rosenbloom et al., 2013*; (*Appendix 2—figure 1B*)). Using this method, the connectivity between proteins with differential gene expression was maximized by the addition of connecting elements, and the model reflects all possible known interactions influenced by the observed differential gene expression. Hierarchy of network modules is known to be related to functional importance (*Szalay-Beko et al., 2012*; *Žitnik et al., 2013*; *Cheng et al., 2015*). The related overlapping modules of interacting proteins (*n* = 64) were then defined within the network model and ranked by their network centrality score (*Appendix 2—figure 1C*). The differential gene expression within the central core (10 genes/proteins) of each of the top 25 network modules was visualized as a transcriptome grid diagram (*Appendix 2—figure 1D*) and used as the basis for further analysis. The central genes of each network module were all shown to have relatively increased connectivity compared to the whole human interactome (*Supplementary file 1i*), suggesting functional relevance and confirming network robustness.

The regulation of gene expression within the transcriptome network model was assessed by defining causal relationships in the known literature up to three links away from each network element with differential expression (*Supplementary file 1j*). The regulatory action of the GC receptor gene, *NR3C1*, was confirmed in relation to the DEGs (p=6.98 × 10$^{-4}$) (*Supplementary file 1j*). Genes identified in this way were mapped to the transcriptome grid diagram and shown to cluster within particular network modules (*Appendix 2—figure 1D*). In addition, a range of DEGs (*FKBP5*, *ZBTB16*, *IGF1R*, *PER1*, *TSC22D3*, and *NCOR2*) were shown to have evidence of *NR3C1* binding to associated regulatory DNA elements. This data was derived from the ENCODE database using a range of cell lines in response to a dexamethasone dose–response (*Davis et al., 2018*; *Casper et al., 2018*).

Within the human interactome model (BioGrid 3.5.178, *n* = 23,273), the GC-regulated genes were identified in both tissues (3.9 × 10$^{-8}$ < p<0.05) and had greater network connectivity than all genes that were not differentially expressed (4.4-fold increase for PBMCs [n = 4,426,426] and 3.7-fold increase for adipose tissue [n = 3,520,520], both p<1 × 10$^{-15}$; Wilcoxon rank-sum test). The enrichment of the connectivity of DEGs demonstrates their functional significance. We interpreted the association between the network properties of these genes and GC action as being indicative of a functional biological relationship. There was very limited overlap of the GC-responsive transcriptome between PBMC and adipose tissue samples (five genes highlighted in Appendix 1).

The gene sets identified as being differentially expressed in response to GC treatment in PBMCs and adipose tissue were shown to be still enriched for connectivity in the human interactome (BioGrid 3.5.178) when compared to 10,000 permutations of the same number of randomly selected genes (3.3-fold increase for PBMCs and 3.2-fold increase for adipose tissue, both comparisons p<1 × 10$^{-15}$, Wilcoxon rank-sum test). This observation showed that these genes have a higher connectivity in the human interactome than expected by chance.

Differential expression in both tissues was indirectly associated with TRIM63-, CALCR-, RHOA-, APOA2-, and ALPL-mediated response to GCs (shared gray circles in *Appendix 2—figure 2A, B*). In this way, DEGs in both tissues were associated with a coherent network related to the gene ontology term 'response to GCs' (*Appendix 2—figure 1A, B*), indicating that different genes are affecting similar pathways in response to GCs. These observations highlight that, while different genes may be involved in the transcriptomic response to GCs in different tissues, there are common pathways through which GC action is manifested. Further evidence for this point was obtained from the overlap of biological pathways associated with GC exposure in PBMCs and adipose tissue

transcriptomes (*Appendix 2—figure 3A, B*). An enriched overlap of 32 biological pathways was identified (2.1-fold enrichment, hypergeometric p<1 × 10⁻⁵).

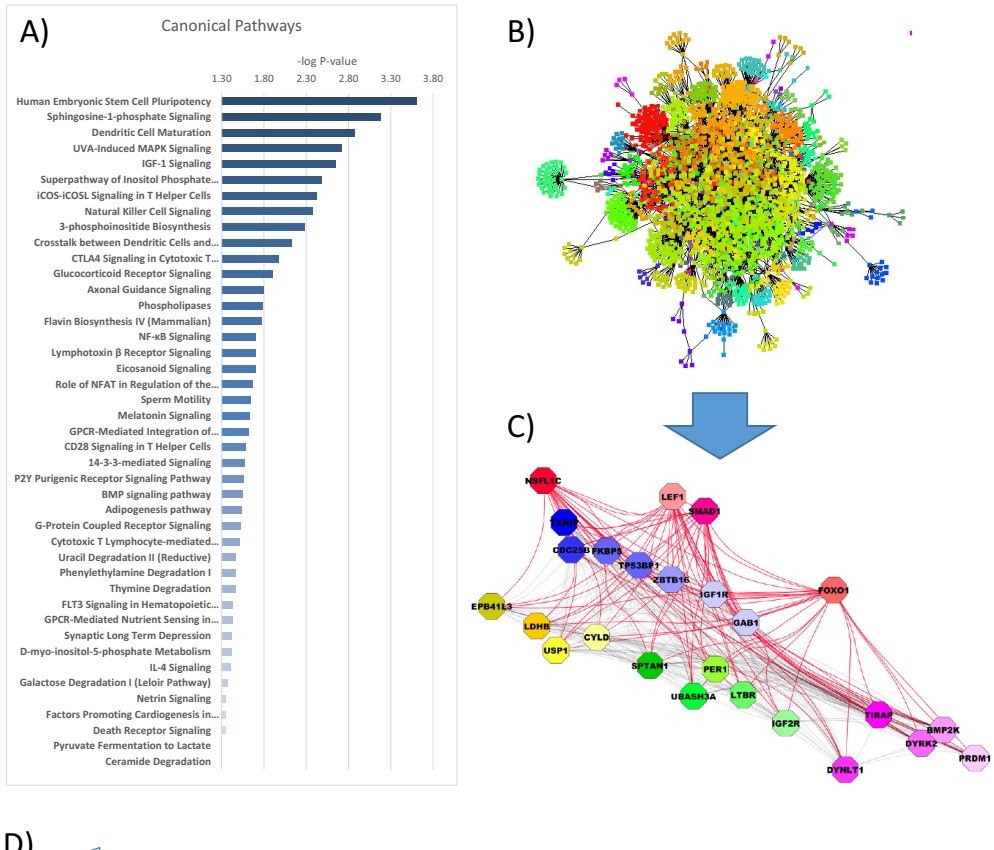

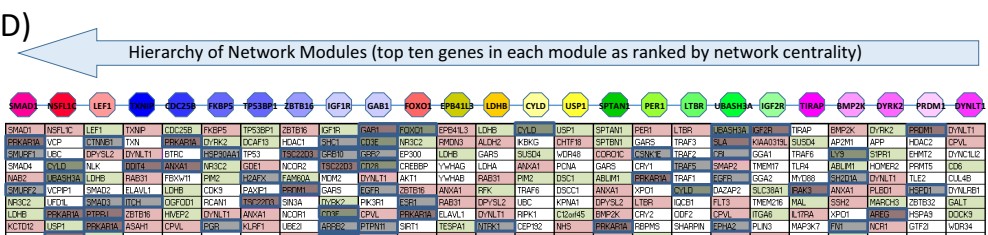

**Appendix 2—figure 1.** Gene ontology and interactome network model of differential gene expression in peripheral blood mononuclear cells (PBMCs) between glucocorticoid (GC) exposure and GC withdrawal. Differentially expressed transcriptomic data in PBMCs was used to determine (**A**) the canonical pathways involved and (**B**) infer an interactome network model using BioGRID database, consisting of 2467 nodes and 64 modules. (**C**) Modules of genes (octagons) within the interactome network model were determined using the Moduland algorithm and ranked by a hierarchy based on network centrality; central modules are functionally relevant as they can influence the activity of a wider proportion of the network. The modules are arranged (left to right) in columns of the top 10 genes in each ranked by network centrality (top to bottom). (**D**) Gene expression marked within the network model (red: up; green: down in hydrocortisone treatment). Causal elements (known literature associations implying regulatory function) are marked in blue.

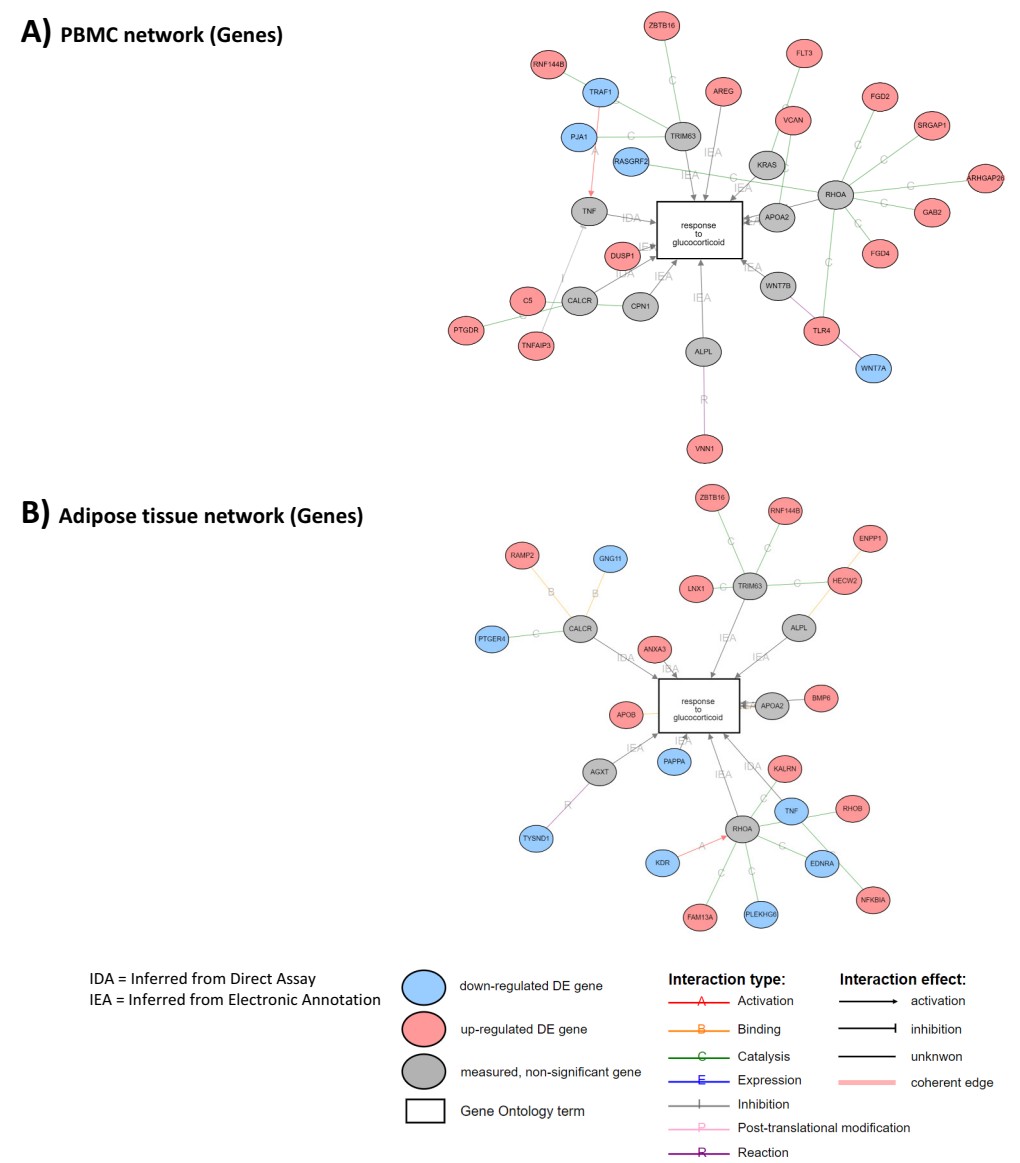

**Appendix 2—figure 2.** Correlated networks related to glucocorticoid (GC) action in the peripheral blood mononuclear cell (PBMC) and adipose tissue transcriptome. The analyses were performed to identify similarities and differences in transcriptome (gene expression) associated to GC exposure in two different tissues. Network associations of GC response to genes with differential expression between GC exposure and GC withdrawal in (**A**) PBMC transcriptome and (**B**) adipose tissue transcriptome. Gray circles represent genes involved in response to GCs that are not differentially expressed. Sixteen genes up-regulated and four down-regulated in PBMCs. Thirteen genes up-regulated and eight down-regulated in adipose tissue.

**A)** Overlapping biological pathways (Gene Ontology)

PBMC    Adipose tissue

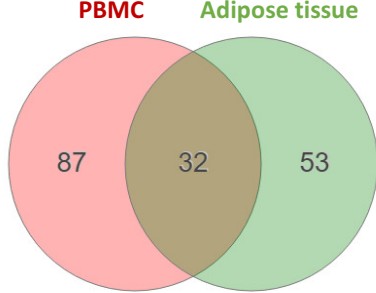

**B)** 32 Overlapping Pathways (Gene Ontology)

| Pathway | PBMC p-value | Adipose Tissue p-value |
|---|---|---|
| Pathways in cancer | 7.6E-07 | 9.4E-04 |
| PI3K-Akt signaling pathway | 5.3E-06 | 3.8E-02 |
| MAPK signaling pathway | 6.1E-05 | 3.7E-04 |
| Phospholipase D signaling pathway | 1.0E-04 | 2.8E-03 |
| Human T-cell leukemia virus 1 infection | 1.1E-04 | 2.7E-02 |
| PD-L1 expression and PD-1 checkpoint pathway in cancer | 1.6E-04 | 3.0E-02 |
| Osteoclast differentiation | 2.3E-04 | 4.5E-02 |
| Human immunodeficiency virus 1 infection | 3.8E-04 | 9.3E-03 |
| Thyroid hormone signaling pathway | 2.3E-03 | 2.4E-02 |
| Wnt signaling pathway | 2.6E-03 | 3.4E-02 |
| Cellular senescence | 2.6E-03 | 1.1E-02 |
| Fc gamma R-mediated phagocytosis | 3.8E-03 | 2.4E-02 |
| B cell receptor signaling pathway | 7.5E-03 | 2.3E-04 |
| Prostate cancer | 7.8E-03 | 1.8E-02 |
| Longevity regulating pathway | 8.8E-03 | 3.4E-02 |
| Kaposi sarcoma-associated herpesvirus infection | 9.5E-03 | 4.7E-02 |
| Inflammatory mediator regulation of TRP channels | 9.6E-03 | 4.2E-03 |
| Sphingolipid signaling pathway | 1.1E-02 | 8.8E-04 |
| Breast cancer | 1.2E-02 | 2.1E-03 |
| EGFR tyrosine kinase inhibitor resistance | 1.4E-02 | 1.4E-02 |
| Central carbon metabolism in cancer | 1.8E-02 | 6.3E-03 |
| Fc epsilon RI signaling pathway | 2.1E-02 | 3.2E-05 |
| Hedgehog signaling pathway | 2.3E-02 | 8.9E-03 |
| Chagas disease (American trypanosomiasis) | 2.4E-02 | 3.2E-02 |
| Prolactin signaling pathway | 2.6E-02 | 1.0E-02 |
| Serotonergic synapse | 2.8E-02 | 4.9E-02 |
| MicroRNAs in cancer | 3.1E-02 | 3.9E-02 |
| T cell receptor signaling pathway | 3.3E-02 | 1.9E-03 |
| FoxO signaling pathway | 3.4E-02 | 5.8E-06 |
| Gastric cancer | 3.9E-02 | 1.1E-02 |
| Cholinergic synapse | 4.6E-02 | 3.6E-03 |
| Long-term depression | 4.6E-02 | 4.4E-02 |

**Appendix 2—figure 3.** Gene Ontology related to glucocorticoid (GC) action in the peripheral blood mononuclear cell (PBMC) and adipose tissue transcriptome. (**A**) Numeric summary of biological pathways (Gene Ontology) associated with differential gene expression between GC exposure and GC withdrawal in the PBMC and adipose tissue transcriptomes. (**B**) Biological pathways (Gene Ontology) associated with differential gene expression in both PBMCs and adipose tissue. Data demonstrate that while different genes may be involved in the transcriptomic response to GCs in the two independent tissues, there are common pathways through which GC action is manifested.

## Appendix 3

### Supplementary materials and methods

#### Plasma cortisol and cortisone

Plasma cortisol and cortisone were analyzed using LC-MS (ABSciex Qtrap 5500 with Waters Acquity UPLC with ACE Excel C18-AR 2.1 × 150 mm column), and urinary-free cortisol and cortisone were analyzed using GC-MS (Thermo TSQ Quantum LC with Trace GC Ultra with Agilent DB17MS 30 m × 0.25 mm × 0.25 µm column) at the Mass Spectrometry Core Laboratory, Centre for Cardiovascular Science, University of Edinburgh, Edinburgh, UK.

#### Isolation of PBMCs

PBMCs were isolated on-site from whole blood using a gradient-based separation procedure and Ficoll-Paque PREMIUM (GE Healthcare). Purified PBMCs were lysed using QIAzol Lysis Buffer (Qiagen, Hilden, Germany) on a QIAshredder column (Qiagen). The lysate was subsequently frozen and stored at –70°C. Samples were eluted in RNAse-free water. RNA concentration was measured spectrophotometrically, and the A260/A280 ratio was 1.97–2.05.

#### Gene expression in PBMCs and adipose tissue

Microarray gene expression analysis in both PBMCs and adipose tissue was performed at the Array and Analysis Facility, Science for Life Laboratory at Uppsala Biomedical Center (BMC), Sweden. The platforms used for molecular profiling were Affymetrix Human Exon 1.0 ST array for PBMCs and 1.1 ST array for adipose tissue. RNA quality was evaluated using the Agilent 2100 Bioanalyzer system (Agilent Technologies, Inc, Palo Alto, CA). Total RNA (150 ng from each PBMC sample and 10 ng from each adipose tissue sample) were used to generate amplified and biotinylated sense-strand cDNA from the entire expressed genome according to the GeneChip WT PLUS Reagent Kit User Manual (P/N 703174 Rev. 1, Affymetrix Inc, Santa Clara, CA). GeneChip ST Arrays (GeneChip Human Gene 2.0 ST Array) were hybridized for 16 hr in a 45°C incubator and rotated at 60 rpm. According to the GeneChip Expression Wash, Stain and Scan Manual (P/N 702731 Rev. 3, Affymetrix Inc), the arrays were then washed and stained using the GeneChip Fluidics Station 450 and finally scanned using the GeneChip Scanner 3000 7G.

#### miRNA analyses

The untargeted miRNA analyses in plasma were performed at Exiqon Services, Denmark. Total RNA was extracted from serum using the miRCURY RNA Isolation Kit-Biofluids (Exiqon, Vedbaek, Denmark). RNA (10 µL) was reverse transcribed in 50 µL reactions using the miRCURY LNA Universal RT microRNA PCR, Polyadenylation, and cDNA Synthesis Kit (Exiqon). cDNA was diluted 50× and assayed in 10 µL PCR reactions according to the protocol for miRCURY LNA Universal RT microRNA PCR; each miRNA was assayed once by qPCR on the microRNA Ready-to-Use PCR, Human Panel I using ExiLENT SYBR Green master mix. Negative controls excluding template from the reverse transcription reaction were treated and profiled like the samples. The amplification was performed in a LightCycler 480 Real-Time PCR System (Roche) in 384-well plates. The amplification curves were analyzed using the Roche LC software, both for determination of Cq (by the second derivative method) and melting-curve analysis.

The targeted miRNA analyses in plasma (including the replication samples) were performed at Exiqon Services, Denmark, at a later date than the untargeted analysis. Total RNA was extracted from the samples using miRCURY RNA Isolation Kit-Biofluids; high-throughput bead-based protocol v.1 (Exiqon, Vedbaek, Denmark) in an automated 96-well format. RNA 2 µL was reverse transcribed in 10 µL reactions using the miRCURY LNA Universal RT miRNA PCR Polyadenylation and cDNA Synthesis Kit (Exiqon). cDNA was diluted 50× and assayed in 10 µL PCR reactions according to the protocol for miRCURY LNA Universal RT miRNA PCR; each miRNA was assayed once by qPCR on the mRNA Ready-to-Use PCR Pick and Mix using ExiLENT SYBR Green master mix. The rest of this analysis was identical with that described above for the untargeted plasma miRNA analysis.

For the miRNA analysis in plasma, the amplification efficiency was calculated using algorithms similar to the LinReg software. All assays were inspected for distinct melting curves and the melting temperature ($T_m$) was checked to be within known specifications for the assay. Furthermore, assays must be detected with 5 Cq less than the negative control, and with Cq < 37 to be included in the data analysis. Data that did not pass these criteria was omitted from any further analysis. Cq was calculated as the second derivative. Using NormFinder, the best normalizer was found to be the average of assays detected in all samples. All data was normalized to the average of assays detected in all samples (average-assay Cq).

## Metabolic profiling of serum by GC-MS and LC-MS

Metabolic profiling of serum by GC-MS and LC-MS was performed at the Swedish Metabolomics Center in Umeå, Sweden.

*Solvents*: Methanol HPLC grade was obtained from Fischer Scientific (Waltham, MA), chloroform Suprasolv for GC from Merck (Darmstadt, Germany), acetonitrile HPLC grade from Fischer Scientific, 2-propanol HPLC grade from VWR (Radnor, PA), and $H_2O$ from Milli-Q (Merck).

*Reference and tuning standards*: Purine 4 µmol/L, HP-0921 [hexakis (1H,1H,3H-tetrafluoropropoxy)phosphazene] 1 µmol/L, Calibrant, ESI-TOF, ESI-L Low Concentration Tuning Mix, and HP-0321 (hexamethoxyphosphazene) 0.1 mmol/L were obtained from Agilent Technologies (Santa Clara, CA).

*Stable isotope internal standards*: LC-MS internal standards: $^{13}C9$-phenylalanine, $^{13}C3$-caffeine, D4-cholic acid, D8-arachidonic acid, and $^{13}C9$-caffeic acid were obtained from Sigma (St. Louis, MO). GC-MS internal standards: L-proline-$^{13}C5$, alpha-ketoglutarate-$^{13}C4$, myristic acid-$^{13}C3$, and cholesterol-D7 were obtained from Cil (Andover, MA); and succinic acid-D4, salicylic acid-D6, L-glutamic acid-$^{13}C5,^{15}N$, putrescine-D4, hexadecanoic acid-$^{13}C4$, D-glucose-$^{13}C6$, and D-sucrose-$^{13}C12$ were obtained from Sigma.

Sample preparation was performed as previously described (*Alwashih et al., 2017a*). A designed randomized run order was made in order to minimize systematic variations within individuals and between time points and treatments. The samples were analyzed according to the designed run order on both GC-MS and LC-MS; GC-MS analysis, derivatization, and GC-MS analysis were performed as described previously (*Jiya et al., 2005*).

For the GC-MS data in serum, all non-processed MS-files from the metabolic analysis were exported from the ChromaTOF software in NetCDF format to MATLAB R2016a (Mathworks, Natick, MA), where all data pre-treatment procedures, such as baseline correction, chromatogram alignment, data compression, and Multivariate Curve Resolution, were performed (*Jonsson et al., 2005*). The extracted mass spectra were identified by comparisons of their retention index and mass spectra with libraries of retention time indices and mass spectra (*Schauer et al., 2005*). Mass spectra and retention index comparison was performed using NIST MS 2.0 software. Annotation of mass spectra was based on reverse and forward searches in the library. Masses and ratio between masses indicative for a derivatized metabolite were especially notified. If the mass spectrum according to SMC's experience was with highest probability indicative of a metabolite and the retention index between the sample and library for the suggested metabolite was ±5 (usually < 3), the deconvoluted 'peak' was annotated as an identification of a metabolite.

For the metabolic profiling of serum by LC-MS, the sample was resuspended in 10 + 10 µL methanol and water. The set of samples was first analyzed in positive mode. After all samples had been analyzed, the instrument was switched to negative mode and a second injection of each sample was performed.

The chromatographic separation was performed on an Agilent 1290 Infinity UHPLC-system (Agilent Technologies, Waldbronn, Germany). A sample (2 µL) was injected onto an Acquity UPLC HSS T3, 2.1 × 50 mm, 1.8 µm C18 column in combination with a 2.1 mm × 5 mm, 1.8 µm VanGuard precolumn (Waters Corporation, Milford, MA) held at 40°C. The gradient elution buffers were (i) $H_2O$, 0.1% formic acid and (ii) 75/25 acetonitrile:2-propanol, 0.1% formic acid with flow rate set at 0.5 mL/min. The compounds were eluted with a linear gradient consisting of 0.1–10% B over 2 min, B was increased to 99% over 5 min and held at 99% for 2 min; B was decreased to 0.1% for 0.3 min and the flow rate was increased to 0.8 mL/min for 0.5 min; and these conditions were held for 0.9 min, after which the flow rate was reduced to 0.5 mL/min for 0.1 min before the next injection.

The compounds were detected with an Agilent 6550 Q-TOF mass spectrometer equipped with a jet stream electrospray ion source operating in positive or negative ion mode. The settings were kept identical between the modes with the exception of the capillary voltage. A reference interface was connected for accurate mass measurements: the reference ions purine (4 µmol/L) and HP-0921 (hexakis(1H,1H,3H-tetrafluoropropoxy)phosphazene) (1 µmol/L) were infused directly into the MS at a flow rate of 0.05 mL/min for internal calibration, and the monitored ions were purine m/z 121.05 and m/z 119.03632; HP-0921 m/z 922.0098 and m/z 966.000725 for positive and negative mode, respectively. The gas temperature was set to 150℃, the drying gas flow to 16 L/min, and the nebulizer pressure to 35 psig. The sheath gas temperature was set to 350℃ and the sheath gas flow to 11 L/min. The capillary voltage was set to 4000 V in positive ion mode and 4000 V in negative ion mode. The nozzle voltage was 300 V. The fragmentor voltage was 380 V, the skimmer 45 V, and the OCT 1 RF Vpp 750 V. The collision energy was set to 0 V. The m/z range was 70–1700, and data was collected in centroid mode with an acquisition rate of 4 scans $s^{-1}$ (1977 transients/spectrum).

For the LC-MS data in serum, all data processing was performed using the Agilent Masshunter Profinder version B.08.00 (Agilent Technologies, Inc). The processing was performed both in a target and an untargeted fashion. For target processing, a predefined list of metabolites commonly found in plasma and serum was searched for using the Batch Targeted feature extraction in Masshunter Profinder. An in-house LC-MS library built up by authentic standards run on the same system with the same chromatographic and mass-spectrometry settings was used for the targeted processing. The identification of the metabolites was based on MS, MS-MS, and retention-time information. For the untargeted data, the pooled quality control samples were processed using Batch Recursive Feature Extraction algorithm within Masshunter Profinder. After exporting cef-files of all processed quality control samples, the Extracted features were matched using Mass Profiler Professional 13.0 (Agilent Technologies, Inc), resulting in a combined recursion file. The recursion file was imported back into Masshunter Profinder and used for Batch Targeted Feature Extraction on all samples.

