## [Decision Letter]

**Acceptance summary:**

The authors have identified candidate markers of glucocorticoid action that can now be investigated in larger cohorts. This work has generated a vast amount of data, including transcriptomics data from both adipose tissue and peripheral blood mononuclear cells, plasma micro RNA data and serum metabolomics data.

**Decision letter after peer review:**

Thank you for submitting your article "Identification of human glucocorticoid response markers using integrated multi-omic analysis" for consideration by *eLife*. Your article has been reviewed by three peer reviewers, and the evaluation has been overseen by a Reviewing Editor and Clifford Rosen as the Senior Editor. The following individual involved in review of your submission has agreed to reveal their identity: Natalie Bordag (Reviewer #2).

The reviewers have discussed the reviews with one another and the Reviewing Editor has drafted this decision to help you prepare a revised submission.

Summary:

The research focuses on the identification of markers of glucocorticoid action both at the transcriptomics level and the metabolite level. The availability of robust markers would be valuable in allowing the optimisation of glucocorticoid dosage. Ten subjects with primary adrenal insufficiency were used to enable biomarker discovery following treatment with hydrocortisone. While the cohort was small many of the markers observed had a high level of significance making this a useful preliminary study. A micro RNA (MiR-122-5p) was identified as being strongly correlated with glucocorticoid exposure and this finding was replicated in independent studies.

Essential revisions:

– Molecular data are often very noisy and require extensive cleaning. Which pre-processing approaches were applied? PCA is normally used to explore the sample structure and identify possible abnormalities while normalisation is needed to rescale the expression values to reduce noise. More explanations are needed on how PCA was used and whether any array normalisation was conducted.

– Certain results are not well explained. For example, those reported in the subsection “PBMC and adipose tissue transcriptomes have limited overlap in response to GC but are enriched for shared pathways” for PBMCs (the same applies to adipose tissue), how many differentially expressed genes (DEGs) did you find? This number is important to know before using those genes in pathway analysis. Did you use FDR or other forms of multiple testing correction to determine significance? It is unclear if you then used those DEGs to test pathway enrichment and you found a connectivity of n=4426 which is 3.7-fold stronger than using random genes. I am not sure what this means, or maybe 4426 is the list of DEGs? If so what is 3.7 connectivity? Please explain. Again, for pathways or GO terms enrichment, FDR should be used to correct for multiple testing. Overall, this analysis (Figure 2) does not appear to add much to the manuscript. Without multiple testing correction, relevant analyses seem to be those based on further gene filtering (i.e. retaining only genes confirmed across tissues and platforms such as those in Figures 4-8).

– Another example of low clarity is Figure 3. What is shared correlation? Heatmaps have no contrast and are uninformative. The dichotomisation by 0.5 is inappropriate, a higher cut-off (e.g. 0.8) might increase the contrast. Maybe the Venn diagram alone (Figure 3E) would be enough to convey your message if better explained?

– Discussion. miR-122-5p is not a novel circulating miRNA, it has been widely monitored as a diagnostic marker. Rephrase.

– There is a very extensive literature on miR-122-5p as diagnostic marker which I don't think has been thoroughly covered in the Discussion. For elevated miR-122-5p has been proposed as predictor on myocardial infarction which in turn has been associated with depressed hydrocortisone levels. Some additional discussion in this direction might be useful.

– Given that many conditions seem to affect miR-122-5p it would be important differentiate the effect of GC exposure from other conditions that affect its levels.

– In the metabolomics data the fold changes are quite small to confident on their robustness in such a small data set. Were the values of the metabolomic markers normally distributed?

---

## [Author Response]

Essential revisions:– Molecular data are often very noisy and require extensive cleaning.

We recognize the point made by the editors/reviewers and have now made clear the extensive work we performed to establish the limits of noise and signal in our multi-omic data sets.

Note that we chose to use multiple approaches to ‘omic data-processing where possible. These approaches are now clearly outlined in the text both in the Materials and methods section and Appendix 1.

Which pre-processing approaches were applied?

This was performed as follows:

a) Transcriptome: RMA (performed using R package oligo) was used to background correct and quantile normalise, prior to summarising to gene-probe set level (subsection “Generation and preparation of ‘omic data” and Appendix 1).

b) Metabolome: Metaboanalyst (via R package MetaboanalystR) was used to filter variables based on ranked interquartile range, normalise metabolites to sample median and log transform the resultant intensities (subsection “Generation and preparation of ‘omic data” and Appendix 1).

c) miRNAome: Qlucore Omics explorer was used to scale and mean centre miRNAome data (subsection “Generation and preparation of ‘omic data”).

PCA is normally used to explore the sample structure and identify possible abnormalities while normalisation is needed to rescale the expression values to reduce noise. More explanations are needed on how PCA was used and whether any array normalisation was conducted.

PCA was performed on all ‘omic data sets to assess the presence of outliers using cross validation. This is now clarified in the Materials and methods section and Appendix 1). No outliers were identified and no batch correction was required as the experimental work was carried out in a single batch.

– Certain results are not well explained.

We have clarified all experimental results, provided extra detail and added extra explanation in the revised manuscript. The implementation of hypernetwork analysis to integrate and define relevant findings represents a novel aspect of this work and we have taken the opportunity to develop reader’s understanding of how we have built this analysis pipeline (Results and Materials and methods).

For example, those reported in the subsection “PBMC and adipose tissue transcriptomes have limited overlap in response to GC but are enriched for shared pathways” for PBMCs (the same applies to adipose tissue), how many differentially expressed genes (DEGs) did you find? This number is important to know before using those genes in pathway analysis. Did you use FDR or other forms of multiple testing correction to determine significance?

Candidate transcripts, miRNAs and metabolites used in hypernetwork analysis were identified initially using a supervised method without FDR correction. We subsequently reanalysed these candidates and determined that [3997/4426] PBMC transcripts, [3115/3520] adipose tissue transcripts and [17/38] metabolites were significantly differentially expressed below an FDR corrected p value of 0.05. FDR adjusted differential expression is not required for hypernetwork analysis as this approach is used as a method of refining a set of potentially relevant ‘omic elements in which to test network dynamics in an unsupervised manner with an assessment of robustness added downstream (see later description of hypernetwork analysis). This has been clarified in the text (subsection “Hypernetworks”), along with the additional explanation in the Results section and the addition of Supplementary file 1G, defining FDR corrected p-values for transcripts and metabolites used in hypernetworks.

Appendix 1 interrogates the similarities between PBMC and adipose tissue transcriptomes, using a false discovery rate modified p-value cut-off of <0.05. These lists are different to those used in hypernetwork analysis and p-value distributions that can be found in Supplementary file 1A.

It is unclear if you then used those DEGs to test pathway enrichment and you found a connectivity of n=4426 which is 3.7-fold stronger than using random genes. I am not sure what this means, or maybe 4426 is the list of DEGs? If so what is 3.7 connectivity? Please explain.

This point has been explained further. The analysis provides data showing an enrichment in connectivity of the differentially expressed genes in the interactome model and supports functional relevance. We have combined these data with the section from the start of the description of the hypernetwork data. These are minor pieces of circumstantial evidence and have now been added to Appendix 2.

Again, for pathways or GO terms enrichment, FDR should be used to correct for multiple testing. Overall, this analysis (Figure 2) does not appear to add much to the manuscript. Without multiple testing correction, relevant analyses seem to be those based on further gene filtering (i.e. retaining only genes confirmed across tissues and platforms such as those in Figures 4-8).

We agree that Figure 2 in the previous version of the manuscript does not add much to the manuscript and have added these results to Appendix 2 (Appendix 2—figure 2 and 3). As we were comparing the same pathways in both PBMC and adipose tissue we have reported p-value rather than FDR to facilitate identification of possible overlap. The data presented in the old Figure 2 highlight the hypothesis that there is a shared relationship in both PBMCs and adipose tissue with GC regulation and as such is minor point. We agree with the reviewer that the filtered analysis presented later is more important for interpretation of the main message of the manuscript.

– Another example of low clarity is Figure 3. What is shared correlation? Heatmaps have no contrast and are uninformative. The dichotomisation by 0.5 is inappropriate, a higher cut-off (e.g. 0.8) might increase the contrast. Maybe the Venn diagram alone (Figure 3E) would be enough to convey your message if better explained?

We have changed all heatmaps to a blue/yellow contrasting colour scheme in alignment with best practise.

We recognise that the hypernetwork concept is a novel concept and have taken the opportunity to spell out the interpretation in the Results section that describes the integrated ‘omic findings.

We now provide a figure to highlight the structure of a hypernetwork and describe more precisely how the analysis works (Figure 3A and B). We specifically explain shared correlation and its relationship to the underlying similarity analysis, although we leave the more precise mathematical description of the matrix structures for explanation in the Materials and methods section. We also provide further clarity in the Materials and methods section on how this method uses the correlation distance to simply define similarity based on the assessment of values >|1.5| of the standard deviation of the distribution of all r-values. The correlation distance is calculated in a similar manner to a Pearson correlation coefficient but the interpretation is as a distance metric. Note that we could “improve” the r-values by using Spearman’s correlation coefficient but doing this removes sensitivity to variation in the data and, importantly, loses a direct relationship to the Euclidean distances between the expression values of different genes. We chose this threshold based on the standard deviation equates to a Pearson correlation coefficient of 0.5 as this value maximised the signal window in the hypernetwork analysis ensuring specific analysis downstream. The hypernetwork signal window was assessed experimentally and these data are now presented (Figure 3—figure supplement 4). Using these approaches, we reflect the small size of the study in the absolute magnitude of the r-value in an unbiased manner but leverage the depth of the multi-omic data sets to increase confidence in the findings. With this experimental pathway the analysis is resistant to low r-value thresholds and robustness testing via comparison of the analysis with the complement of the underlying similarity matrix (i.e. the dissimilarity matrix) adds a final level of confidence in the analysis. The use of the dissimilarity matrix of the data allows a subtraction of those interactions defined as dissimilar from those that are similar and results in a highly robust non-statistical assessment of “presence” in the analysis (subsection “Hypernetworks”). We propose that this approach is more robust than permutation analysis that is also a possible method to determine robustness.

I addition, we have now focussed the description of the results on the Venn diagram which has been simplified for clarity as requested.

– Discussion. miR-122-5p is not a novel circulating miRNA, it has been widely monitored as a diagnostic marker. Rephrase.

We agree with the reviewer and the text in the Discussion section is rephrased accordingly.

– There is a very extensive literature on miR-122-5p as diagnostic marker which I don't think has been thoroughly covered in the Discussion. For elevated miR-122-5p has been proposed as predictor on myocardial infarction which in turn has been associated with depressed hydrocortisone levels. Some additional discussion in this direction might be useful.

We thank the reviewer for bringing this to our attention. New findings concerning the role of miR-122-5p in different disorders are added in the Discussion section. Moreover, a discussion that our miRNA finding (miR-122-5p) may be the functional link between unphysiological glucocorticoid exposure and disorders such as type 2 diabetes mellitus, obesity and cardiovascular disease is added in the Discussion section.

– Given that many conditions seem to affect miR-122-5p it would be important differentiate the effect of GC exposure from other conditions that affect its levels.

In conjunction with the response to the previous comment, we have added in the Discussion section a statement about the potential influence of the known conditions affecting miR-122-5p into our interventions.

– In the metabolomics data the fold changes are quite small to confident on their robustness in such a small data set. Were the values of the metabolomic markers normally distributed?

Quality control was done for both transcriptome and metabolome and normality was demonstrated. We now provide these data in the Materials and methods section and Figure 3—figure supplements 2 and 3.